# Connecting concepts in the brain by mapping cortical representations of semantic relations

Yizhen Zhang [1], Kuan Han[1], Robert Worth [2] & Zhongming Liu [1,3,4,5]✉

In the brain, the semantic system is thought to store concepts. However, little is known about how it connects different concepts and infers semantic relations. To address this question, we collected hours of functional magnetic resonance imaging data from human subjects listening to natural stories. We developed a predictive model of the voxel-wise response and further applied it to thousands of new words. Our results suggest that both semantic categories and relations are represented by spatially overlapping cortical patterns, instead of anatomically segregated regions. Semantic relations that reflect conceptual progression from concreteness to abstractness are represented by cortical patterns of activation in the default mode network and deactivation in the frontoparietal attention network. We conclude that the human brain uses distributed networks to encode not only concepts but also relationships between concepts. In particular, the default mode network plays a central role in semantic processing for abstraction of concepts.

[1] Department of Electrical Engineering and Computer Science, University of Michigan, Ann Arbor, MI, USA. [2] Department of Mathematical Sciences, Indiana University–Purdue University, Indianapolis, IN, USA. [3] Department of Biomedical Engineering, University of Michigan, Ann Arbor, MI, USA. [4] Weldon School of Biomedical Engineering, Purdue University, West Lafayette, IN, USA. [5] School of Electrical and Computer Engineering, Purdue University, West Lafayette, IN, USA. ✉email: zmliu@umich.edu

Humans can describe the potentially infinite features of the world and communicate with others using a finite number of words. To make this possible, our brains need to encode semantics[1], infer concepts from experiences[2], relate one concept to another[3,4], and learn new concepts[5]. Central to these cognitive functions is the brain's semantic system[6]. It is spread widely over many regions in the association cortex[7–9], and it also partially overlaps with the default-mode network[10]. Based on piecemeal evidence from brain imaging studies[11,12] and patients with focal lesions[13], individual regions in the semantic system are thought to represent distinct categories or domains of concepts[11,13] grounded in perception, action, and emotion systems[14,15].

However, little is known about how the brain connects concepts and infers semantic relations[16,17]. As concepts are related to one another in the real world, cortical regions that represent concepts are also connected, allowing them to communicate and work together as networks[18]. It is thus likely that the brain represents semantic relations as emerging patterns of network interaction[19]. Moreover, since different types of concepts may express similar relations, it is also possible that the cortical representation of a semantic relation may transcend any specific conceptual domain. Testing these hypotheses requires a comprehensive study of the semantic system as a set of distributed networks, as opposed to a set of isolated regions. Being comprehensive, the study should also survey cortical responses to a sufficiently large number of words from a wide variety of conceptual domains[1], ideally using naturalistic stimuli[20].

Similar to a prior work[1], we developed a predictive model of human functional magnetic resonance imaging (fMRI) responses given >11 h of natural story stimuli. In this model, individual words and their pairwise relationships were both represented as vectors in a continuous semantic space[21], which was learned from a large corpus and was linearly mapped onto the brain's semantic system. Applying this model to thousands of words and hundreds of word pairs, we have demonstrated the distributed cortical representations of semantic categories and semantic relations, respectively. Our results also shed new light on the role of the default-mode network in semantic processing.

## Results

**Word embeddings predicted cortical responses to speech**. To extract semantic features from words, we used a word2vec model trained to predict the nearby words of every word in large corpora[21]. Through word2vec, we could represent any word as a vector in a 300-dimensional semantic space. Of this vector representation (or word embedding), every dimension encoded a distinct semantic feature learned entirely by data-driven methods[21], instead of by human intuition or linguistic rules[1,22,23]. To relate this semantic space to its cortical representation, we defined a voxel-wise encoding model[24]—a multiple linear regression model that expressed each voxel's response as a weighted sum of semantic features[1] (Fig. 1).

To estimate the voxel-wise encoding model, we acquired whole-brain fMRI data from 19 native English speakers listening to different audio stories (from The Moth Radio Hour: https://themoth.org/radio-hour), each repeated twice for the same subject. We used different stories for different subjects in order to sample more words collectively. We also counterbalanced the stories across subjects, such that the sampled words for every subject covered similar distributions in the semantic space (Supplementary Fig. 1) and included a common set of frequent words (Supplementary Fig. 2 and Supplementary Table 1), while every semantic category or relation of interest was sampled roughly evenly across subjects (Supplementary Figs. 3 and 4). In total, the story stimuli combined across subjects lasted 11 h and included 47,356 words (or 5228 words if duplicates were excluded). The voxel-wise encoding model was estimated based on the fMRI data concatenated across all stories and subjects.

By 10-fold cross-validation[25], the model-predicted response was significantly correlated with the measured fMRI response (block-wise permutation test, false discovery rate or FDR $q<0.05$) for voxels broadly distributed on the cortex (Fig. 2). The voxels highlighted in Fig. 2 were used to delineate an inclusive map of the brain's semantic system, because the cross-validation was applied to a large set of (5228) words, including those most frequently used in daily life (Supplementary Fig. 2). This map, hereafter referred to as the semantic system, was widespread across regions from both hemispheres, as opposed to only the left hemisphere, which has conventionally been thought to dominate language processing and comprehension[26].

We also tested how well the trained encoding model could be generalized to a new story never used for model training and whether it could be used to account for the differential responses at individual regions. For this purpose, we acquired the voxel response to an independent testing story (6 m 53 s, 368 unique words) for every subject and averaged the response across subjects. As shown in Fig. 3a, we found that the encoding model was able to reliably predict the evoked responses in the inferior frontal sulcus (IFS), supramarginal gyrus (SMG), angular gyrus (AG), superior temporal gyrus (STG), middle temporal visual area (MT), left fusiform gyrus (FuG), left parahippocampal gyrus (PhG), and posterior cingulate cortex (PCC) (block-wise permutation test, FDR $q<0.05$). These regions of interest (ROIs), as predefined in the human brainnetome atlas[27] (Fig. 3b, see details in Supplementary Table 2), showed different response dynamics given the same story, suggesting their highly distinctive roles in semantic processing (Fig. 3c). Despite such differences across regions, the encoding model was found to successfully predict the response time series averaged within every ROI except the right FuG (Fig. 3c), suggesting its ability to explain the differential semantic coding (i.e., stimulus–response relationship) at different regions.

**Distributed cortical patterns encoded semantic categories**. Since the encoding model was generalizable to new words and sentences, we further used it to predict cortical responses to >9000 words from nine categories: tool, human, plant, animal, place, communication, emotion, change, quantity (Supplementary Table 3), as defined in WordNet[28] and are representative of different conceptual domains. We confined the model prediction to the voxels in the semantic system for which the model fit was significant during cross-validation (Fig. 2). Within each category, we averaged the model-predicted responses given every word and mapped the statistically significant voxels (one-sample $t$-test, FDR $q < 0.01$). This map represented each category being projected from the semantic space to the cortex, and thus was interpreted as the model-predicted cortical representation of each category. We found that individual categories were represented by spatially overlapping and distributed cortical patterns (Fig. 4). For example, the category tool was represented by the SMG, posterior middle temporal gyrus (pMTG), FuG, and inferior frontal gyrus (IFG); this representation was more pronounced in the left hemisphere than the right hemisphere. Such categories as human, plant, and animal were also represented more by the left hemisphere than the right hemisphere. The category place was represented by bilateral PhG, dorsolateral prefrontal cortex, and AG. In contrast, communication, emotion, change, and quantity, showed stronger representations in the right hemisphere than in the left hemisphere. Although the size of word samples varied

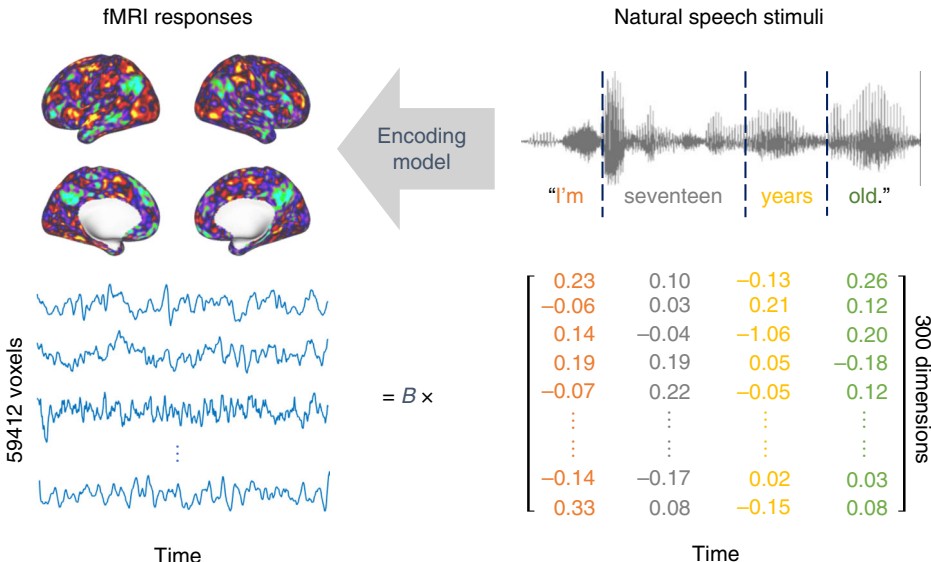

**Fig. 1 Illustration of the encoding model.** The encoding model was trained and tested for predicting the fMRI responses (top left) to a time series of words in audio-story stimuli (top right). Every word (as color coded) was converted to a 300-dimensional vector through word2vec. The encoding model was denoted as a 59,421-by-300 matrix (B) to predict the voxel response to every word (bottom).

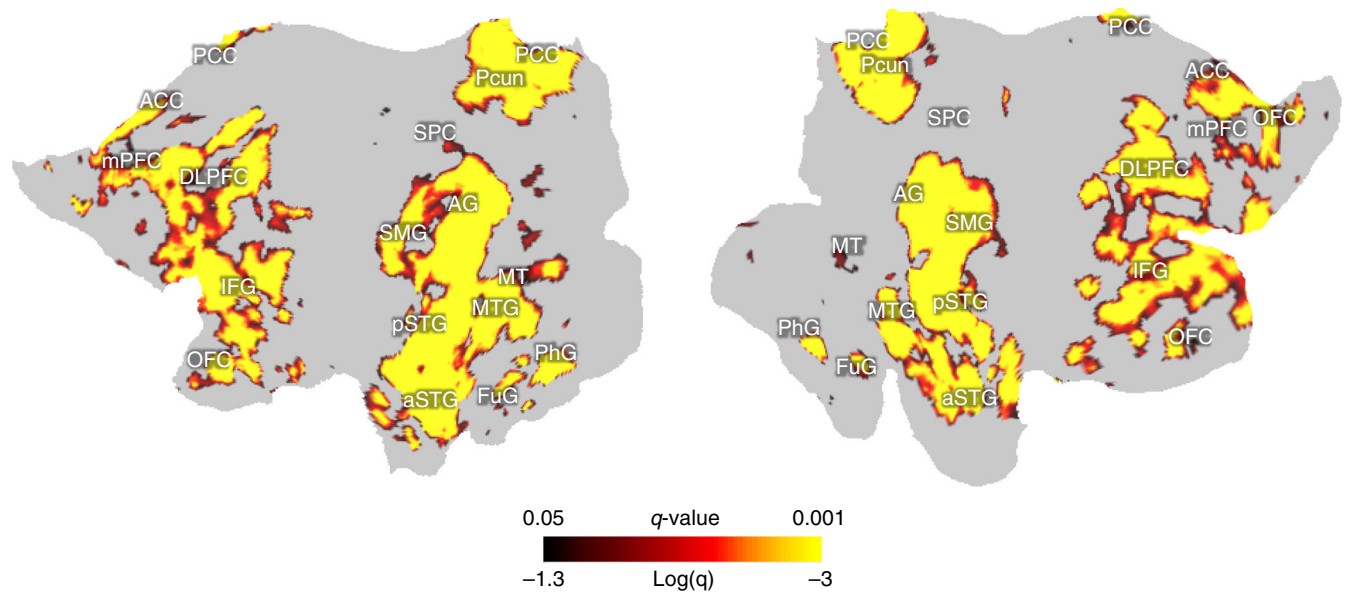

**Fig. 2 A map of the semantic system obtained by 10-fold cross-validation of the encoding model.** The map is displayed on the flattened cortical surfaces for the left and right hemispheres. The color indiciates the FDR (or $q$ value) in a logarithmic scale. The highlighted areas include voxels where cross-validation results are statistically significant (block-wise permutation test, one-sided, $q<0.05$). Anatomical labels are shown to relate the model-predictable areas to cortical regions predefined in a connectivity-based atlas[27].

across categories (Supplementary Table 3), the sample size was sufficiently large for every category, since the resulting category representation had reached or approached its maximum extent at the given sample size (Supplementary Fig. 5). See Supplementary Method 6 for more details about testing the effect of sample size on categorical representation.

To each voxel in the semantic system, we assigned a single category that gave rise to the strongest voxel response, thus dividing the semantic system into category-labeled parcels (Fig. 5a). The resulting parcellation revealed how every category of interest was represented by a different set of regions, as opposed to any single region. In addition, the distinction in left/right lateralization was noticeable and likely attributable to the

varying degree of concreteness for the words from individual categories. The concepts lateralized to the left hemisphere appeared relatively more concrete or exteroceptive, whereas those lateralized to the right hemisphere were more abstract or interoceptive (Fig. 5b). This intuitive interpretation was supported by human rating of concreteness (from 1 to 5) for every word in each category[29]. The concreteness rating was high (between 4 and 5) for the categories lateralized to the left hemisphere, whereas it tended to be lower yet more variable for those categories dominated by the right hemisphere (Fig. 5c).

**Co-occurring activation and deactivation encoded word relation.** Through the word2vec model, we could also represent

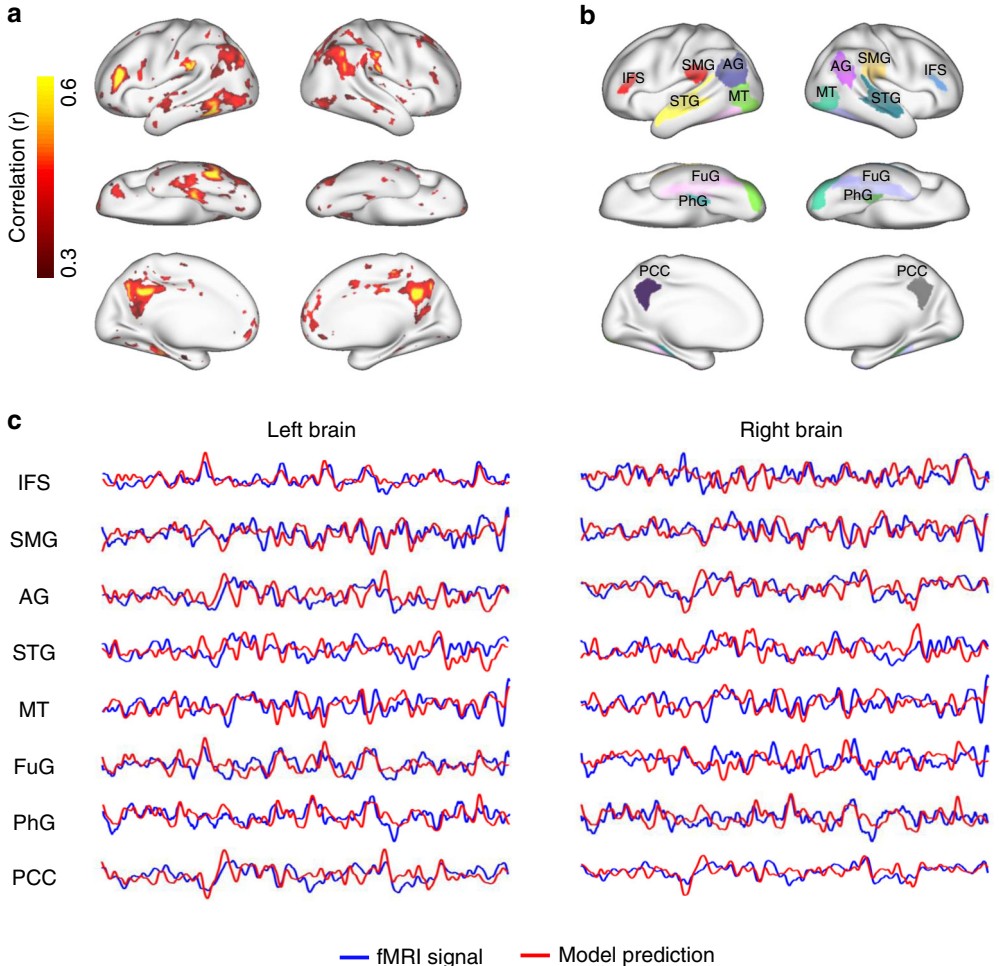

**Fig. 3 Measured vs. model-predicted responses to a new (untrained) testing story. a** The voxel-wise correlation between fMRI responses and model predictions for a 6 m 53 s testing story. The fMRI responses were averaged across subjects. Color indicates the correlation coefficient. The color-highlighted areas include the voxels of statistical significance (block-wise permutation test, one-sided, FDR $q < 0.05$). **b** Predefined ROIs (shown in different colors) are displayed on the cortical surfaces. See details in Supplementary Table 2. **c** Response time series as measured (blue) or model-predicted (red) for each ROI, by averaging the time series across voxels within each ROI. IFS inferior frontal sulcus, SMG supramarginal gyrus, AG angular gyrus, STG superior temporal gyrus, MT middle temporal visual area, FuG fusiform gyrus, PhG parahippocampal gyrus, PCC posterior cingulate gyrus.

semantic relations as vectors in the semantic space[30]. Specifically, we represented the relationship between any pair of words based on their difference vector in word embedding. We chose word pairs from the SemEval-2012 Task 2 dataset[31]. Every chosen word pair had been human rated as an affirmative example of one of ten classes of semantic relation: whole-part, class-inclusion, object-attribute, case relations, space-associated, time-associated, similar, contrast, object-nonattribute, and cause-effect (Supplementary Table 4). For the first six classes, the relation vectors in the semantic space were found to be more consistent across word pairs in the same class than those in different classes (Supplementary Fig. 6). For each of the first six classes, the relation between every pair of words could be better identified based on the relation vectors of the other word pairs in the same class than those from any different class, showing the top-1 identification accuracy from 40% to 83% against the 10% chance level (Supplementary Fig. 7).

For a given word pair, their relation vector could be further projected onto the cortex through the encoding model. For an initial exploration, we applied this analysis to 178 word pairs that all shared a whole-part relationship. For example, in four word pairs, (hand, finger), (zoo, animal), (hour, second), and (bouquet,

flower), finger is part of hand; animal is part of zoo; second is part of hour; flower is part of bouquet. Individually, the words from different pairs had different meanings and belonged to different semantic categories, as finger, animal, second, and flower were semantically irrelevant to one another. Nevertheless, their pairwise relations all entailed the whole-part relation, as illustrated in Fig. 6a. By using the encoding model, we mapped the pairwise word relationship onto voxels in the semantic system (as shown in Fig. 2), averaged the results across pairs, and highlighted the significant voxels (paired permutation test, FDR $q < 0.05$). The resulting cortical map represented each semantic relation being projected from the semantic space to the cortex, reporting the model-predicted cortical representation of that relation. We found that the whole-part relation was represented by a cortical pattern that manifested itself as the co-occurring activation of the DMN[32] (including AG, MTG, and PCC) and deactivation of the frontoparietal network[33,34] (FPN, including LPFC, IPC, and pMTG) (Fig. 6b). This cortical pattern encoded the whole-part relation independent of the cortical representations of the individual words in this relation. The co-activation and deactivation pattern indicated that conceptual progression from part to whole manifested itself as increasing deactivation of

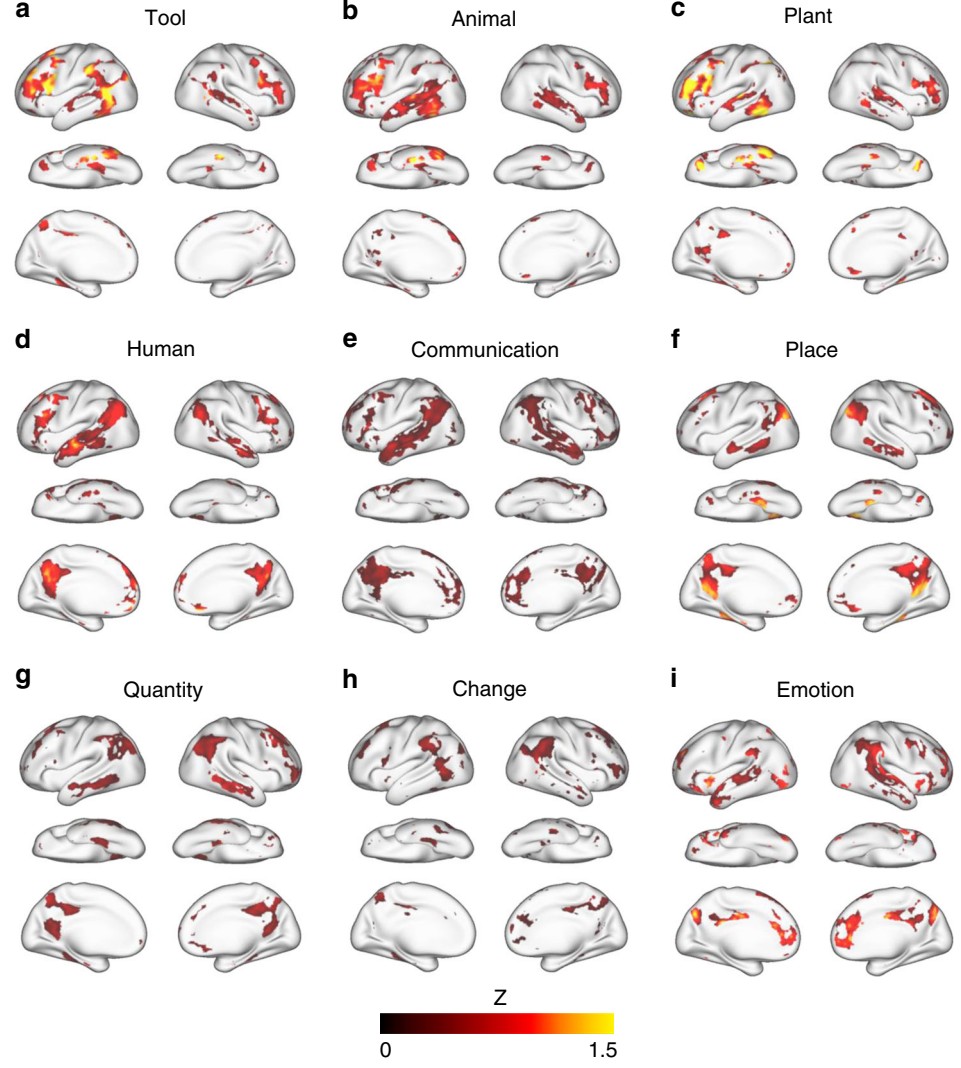

**Fig. 4 Cortical representations of semantic categories.** For each category, the color indicates the mean of the normalized response (or the z score) averaged across word samples in the category (Supplementary Table 3). The color-highlighted areas include the voxels of statistical significance (one-sample t-test, one-sided, FDR q < 0.01).

FPN alongside increasing activation of DMN, whereas progression from whole to part was shown as the reverse cortical pattern varying in the opposite direction, as illustrated in Fig. 6c.

Similarly, we also mapped the cortical representations of several other semantic relations. Each relation was projected to a distinct cortical pattern (Fig. 7). Specifically, the class-inclusion relation, e.g., (color, green) where color includes green, was represented by the activation of AG and MTG and the deactivation of IFG and STG (Fig. 7b). The object-attribute relation, e.g., (fire, hot) where fire is hot, was represented by an asymmetric cortical pattern including activation primarily in the left hemisphere and deactivation primarily in the right hemisphere (Fig. 7c). The case relations, e.g., (coach, player) where a coach teaches a player, was represented by a cortical pattern similar to that of the whole-part relation (Fig. 7d), despite a lack of intuitive connection between the two relations. The space-associated relation, e.g., (library, book) where book is an associated item in a library, was represented by activation of AG and PCC and deactivation of STG (Fig. 7e). Lastly, the time-associated relation, e.g., (morning, sunrise) where sunrise is a phenomenon associated with morning, was also represented by a bilaterally asymmetric pattern (Fig. 7f). A graph-based illustration

of the representational geometry further highlights the distinction across semantic relations in terms of their bilateral (a)symmetry and engagement of individual ROIs (Supplementary Fig. 8). However, several nominal (human-defined) relations, e.g., similar, contrast, object-nonattribute, and cause-effect, were projected onto either no or fewer voxels (Supplementary Table 4 and Supplementary Fig. 9).

The voxel-wise univariate analysis restricted the representation of each semantic relation to one cortical pattern while ignoring the interactions across voxels and regions. This limitation led us to use a principal component analysis (PCA) to decompose the cortical projection of the difference vector of every word pair in each semantic relation. This multivariate analysis revealed two cortical patterns that were statistically significant (one-sample t-test, p < 0.01) in representing the semantic relation of object-attribute, case relations, or space-associated, but only revealed one pattern for the relation of whole-part, class-inclusion, time-associate, or cause-effects (Supplementary Fig. 9). Interestingly, when two cortical patterns represented one class of semantic relation, they seemed to correspond to different subclasses of that relation (Fig. 8). For the relation of object-attribute, one cortical pattern corresponded to inanimate object-attribute, e.g., (candy,

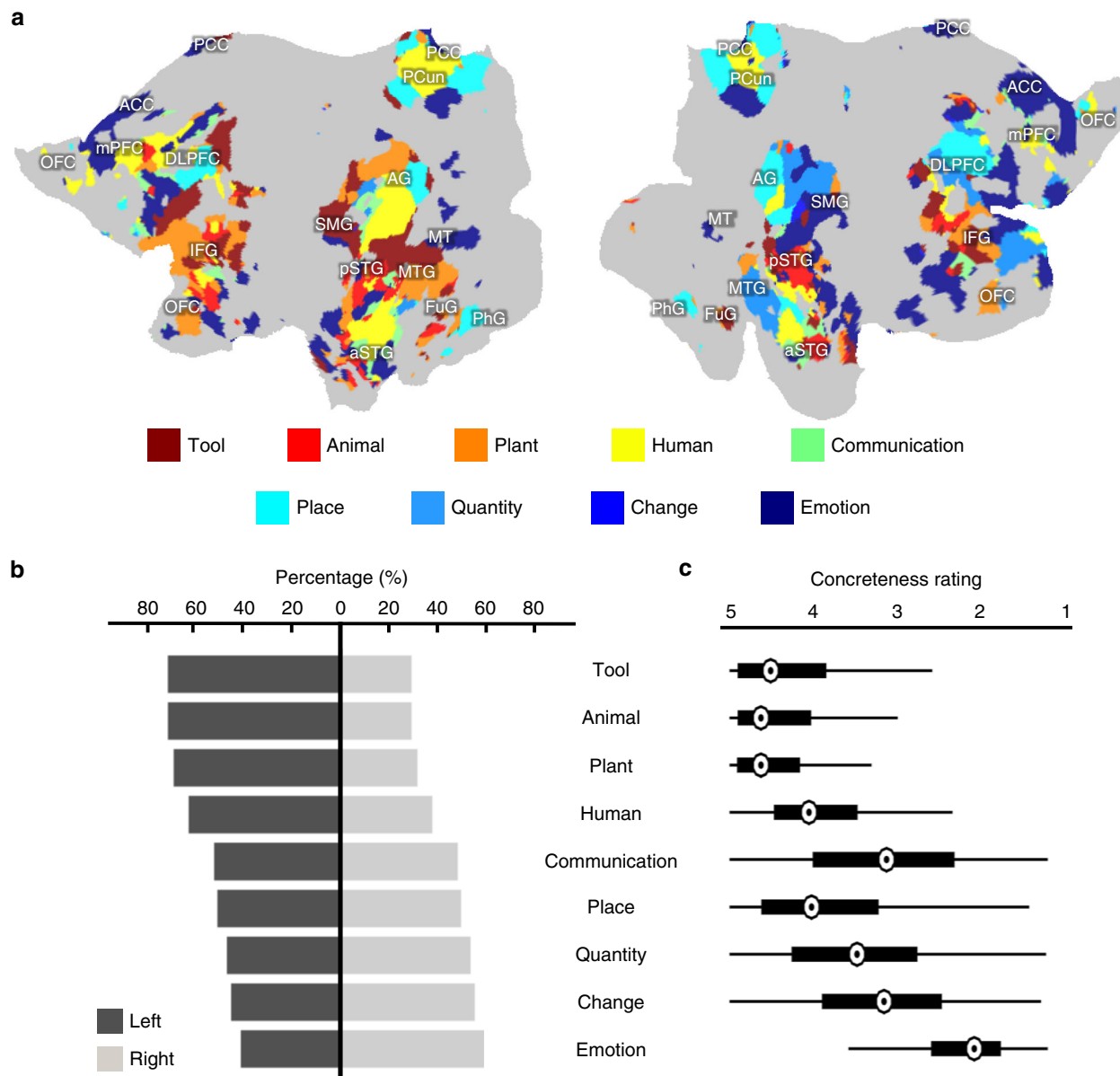

**Fig. 5 Cortical representation of semantic category. a** Category-labeled parcellation based on voxel-wise selectivity using a "winners take all" strategy. **b** Cortical lateralization of categorical representations. For each category, the percentage value was calculated by counting the number of voxels on each hemisphere that represented the given category. The number of word samples in each of the nine categories is: tool (200), animal (734), plant (387), human (808), communication (2027), place (814), quantity (958), change (3417), and emotion (504). **c** The concreteness rating of words in each category. The maximum value of the concreteness rating is 5.00 and the minima value is 1.25. In this box plot, the central mark indicates the median, and the box edges indicate the 25th and 75th percentiles, respectively. The maximum whisker length is 1.5.

sweet), and the other corresponded to human-attribute, e.g., (coward, fear) (Fig. 8a). Similarly, the two cortical patterns for case relations corresponded to agent-instrument and action-recipient, respectively (Fig. 8b). The space-associated relation was distinctively represented for its two subclasses: space-associated item and space-associated activity (Fig. 8c). The cortical patterns that represented a semantic relation, as obtained with either the multivariate or univariate analysis, highlighted generally similar regions (Supplementary Fig. 9).

## Discussion

Using fMRI data from subjects listening to natural story stimuli, we established a predictive model to map the cortical representations of semantic categories and relations. We found that

semantic categories were not represented by segregated cortical regions but instead by distributed and overlapping cortical patterns, mostly involving multimodal association areas. Although both cerebral hemispheres supported semantic representations, the left hemisphere was more selective to concrete concepts, whereas the right hemisphere was more selective to abstract concepts. Importantly, semantic relations were represented by co-occurring activation and deactivation of distinct cortical networks. Semantic relations that reflected conceptual progression from concreteness to abstractness were represented by the co-occurrence of activation in the default-mode network and deactivation in the attention network. Interestingly, some semantic relations could each be represented by two cortical patterns, corresponding to intuitively distinct subclasses of the relation. Our findings suggest that the human brain represents a

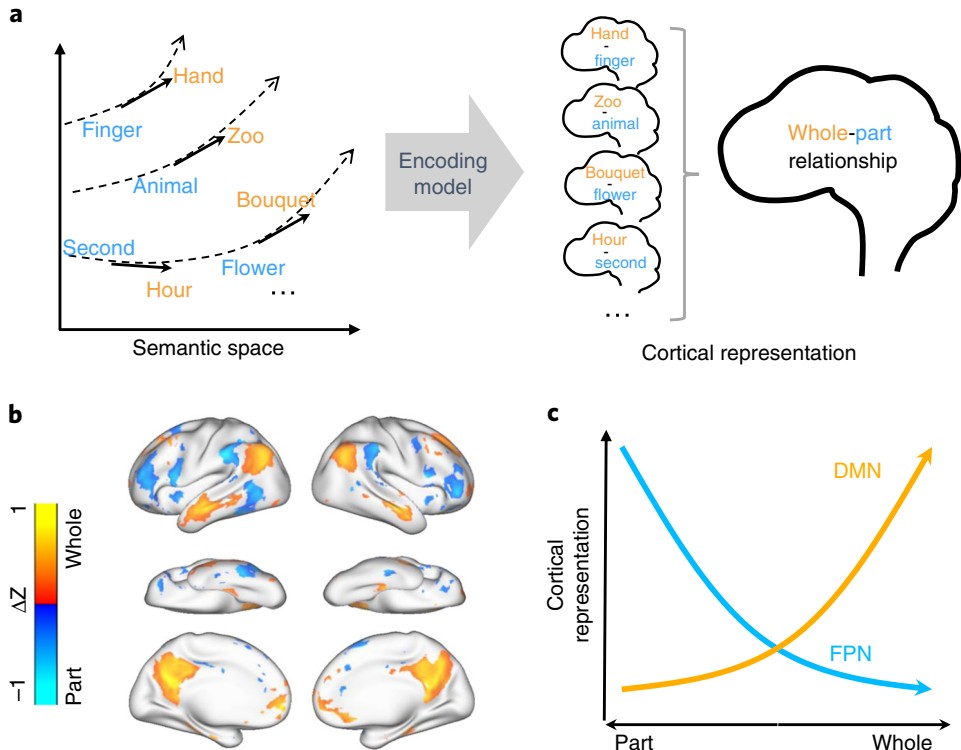

**Fig. 6 Mapping cortical representation of the whole-part relation. a** The illustration of mapping the whole-part relation from the semantic space to the human brain through the voxel-wise encoding model. We viewed the whole-part relation as a vector field over the semantic space. This relation field was sampled by the difference vector of each word pair that held such a relation (left). The cortical representation of this difference vector was predicted by the voxel-wise encoding model. Cortical representation of the whole-part relation was then obtained by averaging representations of all word pairs (right). **b** Cortical representation of the whole-part relation. The statistical significance was assessed by a paired permutation test (178 word pairs, two-sided, FDR $q < 0.05$). **c** The co-occurring activation of DMN and deactivation of FPN encodes the whole-part relation or the conceptual progression from part to whole.

continuous semantic space. To support conceptual inference and reasoning, the brain uses distributed cortical networks to encode not only concepts but also relationships between concepts. Notably, the default-mode network plays an active role in semantic processing for abstraction of concepts. In the following, we discuss our methods and findings from the joint perspective of machine learning and neuroscience in the context of natural language processing.

Central to this study is the notion of embedding concepts in a continuous semantic space[21]. Although we use words to study concepts, words and concepts are different. The vocabulary is finite, but concepts are infinite. We view words vs. concepts as discrete vs. continuous samples from the semantic space. Moving a concept in the semantic space may create a new concept or arrive at a different concept. This provides the flexibility for using concepts to describe the world, as is understood by the brain and, to a lesser extent, as is expressed in language.

Moreover, concepts are not isolated but related to one another. Since we view concepts as points in the semantic space, we consider conceptual relationships to be continuous vector fields in the same space. A position in the semantic space may experience multiple fields, and different positions may experience the same field. Thus, a concept may relate to other concepts in various ways, and different pairs of concepts may hold the same relation[4]. Because semantics reflect cognitive functions that enable humans to understand and describe the world, we hypothesize that the brain not only encodes such a continuous semantic space[1] but also encodes semantic relations as vector fields in the semantic space.

Machine learning leverages the notion of continuous semantic space for natural language processing[21,35,36], and provides a new

way to model and reconstruct neural responses[1,24,37–39]. For example, word2vec can represent millions of words as vectors in a lower-dimensional semantic space[21]. Two aspects of word2vec have motivated us to use it for this study. First, words with similar meanings share similar linguistic contexts and have similar vector representations in the semantic space[40]. Second, the relationship between two words is represented by their difference vector, which is transferable to another word. For an illustrative example, "(man − women) + queen" results in a vector close to "king"[30]. As such, word2vec defines a continuous semantic space and preserves both word meanings and word-to-word relationships.

In addition, word2vec learns the semantic space from large corpora in a data-driven manner[21]. This is different from defining the semantic space based on keywords that are hand selected[22], frequently used[1], minimally grounded[41], or neurobiologically relevant[23,42]. Although those word models are seemingly more intuitive, they are arguably subjective and may not be able to describe the complete semantic space. We prefer word2vec as a model of word embedding, because it leverages big data to learn natural language statistics without any human bias. We assume that the brain encodes a continuous semantic space similarly as is obtained by word2vec. Since word2vec is not constrained by any neurobiological knowledge, we do not expect it to encode the exactly same semantic space as does the brain. Instead, we hypothesize that the word2vec-based semantic space and the brain are similar up to linear projection (i.e., transformation through linear encoding).

Our results support this hypothesis and reveal a distributed semantic system (Fig. 2). In this study, the semantic system mapped with natural stories and thousands of words resembles the semantic system mapped with meta-analysis of the activation

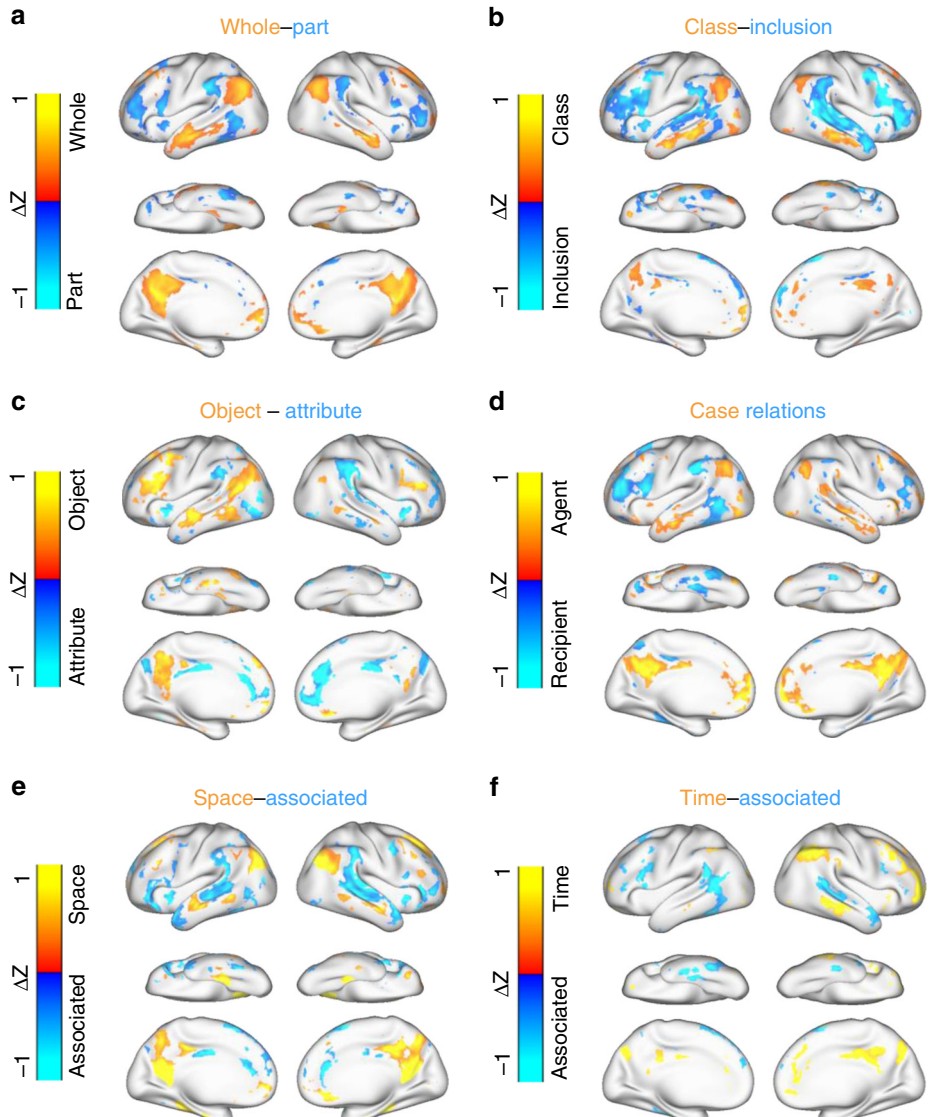

**Fig. 7 Cortical representations of semantic relations.** The cortical pattern associated with each relation shows the average cortical projection of every word-pair sample in that relation and highlights only the voxels of statistical significance (paired permutation test, two-sided, FDR q<0.05) based on voxel-wise univariate analysis. For each of the six relations in this figure, see Supplementary Table 4 for summary statistics and also see Supplementary Fig. 9 for the results about other relations.

foci associated with fewer words[6]. As in that paper, our results also highlight a similar set of semantics-encoded regions (Fig. 2), most of which are associated with high-level integrative processes that transcend any single modality[8,9]. However, our map of the semantic system is largely bilateral, rather than being dominated by the left hemisphere as suggested by Binder et al.[6], although the activation foci analyzed by Binder et al. are actually distributed on both hemispheres (see Fig. 2 in Binder et al.[6]). Importantly, the two hemispheres seem to be selective to different aspects of semantics. Unlike prior findings[43,44], our results suggest that the left hemisphere tends to encode exteroceptive and concrete concepts, whereas the right hemisphere tends to encode inter-oceptive and abstract concepts (Figs. 4 and 5).

Our semantic system shows a cortical pattern similar to that reported by Huth et al.[1]. This similarity is not surprising, because both studies use similar natural speech stimuli and encoding models. However, unlike Huth et al.[1], we do not emphasize the semantic selectivity of each region or tile the cortex into regions associated with distinct conceptual domains. On the contrary,

none of the conceptual categories addressed in this study is represented by a single cortical region. Instead, individual categories are represented by spatially distributed and partly over-lapping cortical networks (Fig. 4), each of which presumably integrates various domain-defining attributes by connecting the regions that encode different attributes[11,14,15]. In this regard, our results lend support to efforts that address semantic selectivity by means of networks, as opposed to regions[18,19].

The primary focus of this study is on semantic relations between words. Extending the earlier discussion about the semantic space, the relationship between words is represented by their vector difference, of which the direction and magnitude indicate different aspects of the relationship. Let us use (minute, day) as an example. Of their relation vector, the direction indicates a part-to-whole relation, and the magnitude indicates the offset along this direction. Starting from minute and relative to day, a larger offset leads toward month or year, a smaller offset leads toward hour, and a negative offset leads toward second. Our results suggest that the direction of relation vector tends to be

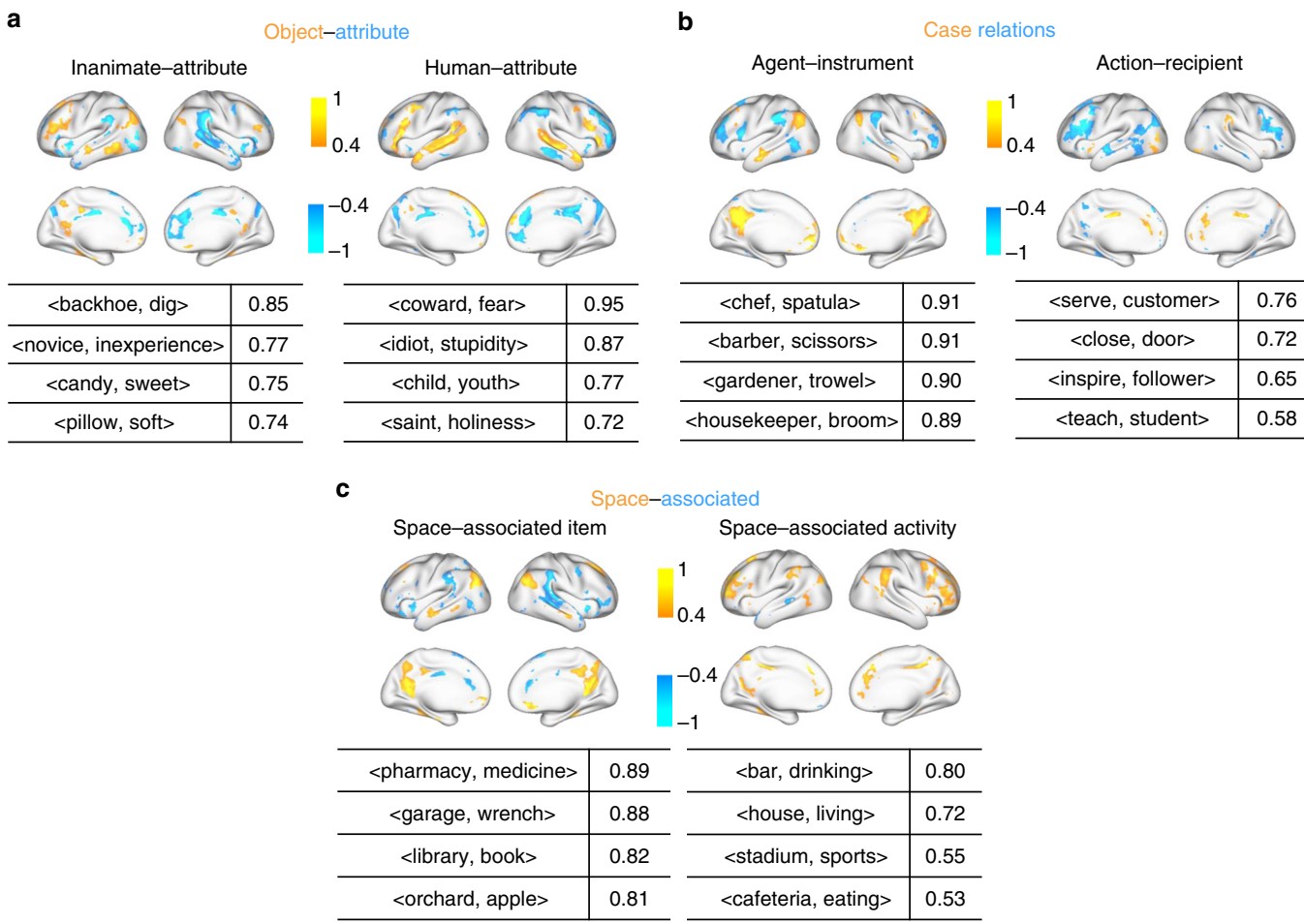

**Fig. 8 Multiple cortical patterns represent subclasses of individual semantic relations.** For three semantic relations: "object-attribute" (**a**), "case relations" (**b**), and "space-associated" (**c**), multivariate pattern analysis revealed two cortical patterns associated with each of these relations. The cortical patterns were min–max normalized to [−1, 1]. The table lists the top-4 word pairs, of which the cortical projection was most similar (in terms of cosine similarity) with the first (left) or second (right) cortical pattern associated with each relation. See Supplementary Fig. 9 for results about other relations.

generalizable and transferrable across word pairs in the same semantic relation (Supplementary Figs. 6 and 7). This leads us to hypothesize that the semantic space includes continuous vector fields, each of which represents a semantic relation and is likely applicable to various concepts or even domains of concepts. When a vector field is visualized as many field lines, the points (i.e., concepts) that each field line passes through are related to one another by the same semantic relation (as illustrated by Fig. 6a).

In a nominal relation (e.g., whole-part), each word pair takes a discrete sample from the underlying vector field (Fig. 6a). Projecting a number of such relational samples onto the cortex reveals one or multiple cortical patterns that encode the relation. Such cortical patterns often manifest themselves as co-occurring activation and deactivation of different regions (Figs. 6, 7, and 8). We interpret this co-activation and co-deactivation as an emerging pattern when the brain relates two concepts that hold a meaningful relation, reflecting the progression from one concept to the other. This pattern encodes generalizable differential relations between concepts, as opposed to concepts themselves, serving differential coding that transcends any conceptual domain or category (Fig. 6). Speculatively, this network-based coding of semantic relation is an important mechanism that supports analogical reasoning[45], e.g., matching similar relations with different word-pair samples[31]. This plausible mechanism of the brain might further facilitate humans learning new concepts

by connecting them to existing concepts through established semantic relations.

It is also noteworthy that a semantic relation as defined by human intuition may not exactly match the relation as represented by the brain. It is possible that a nominal relation may be heterogeneous and contain multiple subclasses each being represented by a distinct cortical pattern. Results obtained with multivariate pattern analysis (MVPA) support this notion (Fig. 8). It is also reasonable that a nonsensical relation, e.g., object-nonattribute, does not have any cortical representation (Supplementary Fig. 9).

Although we view word relations and categories as distinct aspects of the semantic space, the two aspects may engage similar cortical networks under specific circumstances. For example, our results indicate that the space-associated relation and the place category are represented by similar cortical patterns (Figs. 4f and 7e). This is unsurprising because the space-associated word pairs are often associated with place. Such a relation-category association is intrinsic to natural language statistics, and similarly applies to the time-associated relation and the quantity category. This does not imply that the semantic relations are always associated with specific semantic categories (Supplementary Fig. 10). There is no evidence for a generalizable relation-category association. See Supplementary Method 5 for more details about evaluating the association between relations and categories.

We interpret the co-activation/deactivation patterns as "anti-correlated networks" with respect to the cortical representations of semantic relations. This interpretation is reasonable given the notion of "activate together wire together". Task-related patterns of cortical activation resemble those emerging from spontaneous activity or resting state networks[46]. In the context of semantics, the anti-correlated networks reported herein encode a semantic relation, or the direction in which one concept relates to another. For example, conceptual progression from part to whole has a cortical signature as co-occurring activation in DMN and deactivation in FPN (Fig. 6b). The opposite direction from whole to part involves the same regions or networks but reverses their polarity in terms of activation or deactivation. In this example, the cortical co-activation/deactivation pattern is nearly identical to the anti-correlated networks observed with resting state fMRI[47], and therefore it is likely to be intrinsic and supported by underlying structural connections.

Our results suggest that DMN is involved in cortical processing of not only concepts but also semantic relations. This finding underscores the fact that DMN plays an active role in language and cognition[10,48–51], rather than only a task-negative and default mode of brain function[32]. In particular, several semantic relations, such as whole-part and class-inclusion, are all mapped onto DMN (Figs. 6 and 7), suggesting that DMN is likely associated with the semantic regularity common to those relations. Indeed, words being whole or class are more abstract and general, whereas those being part or inclusion are relatively more detailed and specific. As such, these relations all indicate (to a varying degree) conceptual abstraction. This progression involves DMN, increasing or decreasing its activity as a concept (of various types) becomes more abstract or specific, respectively. Moreover, concrete concepts, e.g., tool, plant, animal, are represented by regions outside the DMN, whereas more abstract concepts, e.g., communication, emotion, quantity, are represented by cortical regions that reside in, or at least overlap with, DMN (Figs. 4 and 5).

These observations lead us to speculate that DMN underlies a cognitive process for abstraction of concepts. This interpretation is consistent with findings from several prior studies[51,52]. For example, Spunt et al. have shown that conceptualizing the same action at an increasingly higher level of abstraction gives rise to an increasingly greater responses at regions within DMN[52]. Sormaz et al. have shown evidence that activity patterns in DMN during cognitive tasks are associated with whether thoughts are detailed, rather than whether they are task related or unrelated[51]. In contrast to DMN, another network, FPN, seems to play an opposite role in semantic processing. FPN is often activated by attention-demanding tasks and is intrinsically anti-correlated with DMN[47]. Our results suggest that FPN is increasingly activated when the brain is engaged in conceptual specification.

Although our experimental design is justifiable by practical and methodological considerations, it is worth further noting potential limitations, additional justifications and future directions. In this study, we used different stories for different subjects to collect a large set (47,356 words) of stimulus–response samples for training the encoding model. It is logistically difficult to acquire enough data from a single subject. A typical fMRI experiment lasts <2 h to avoid fatigue, while >11 h of fMRI scans as needed for the desired sample size would be too long to be realistic. This study design might be of potential concern that individual differences, e.g., laterality[26,53], are confounded with the words used for model training. If a subset of subjects is overrepresented for one semantic dimension and a different subset is overrepresented for a different dimension, the trained encoding model would reflect the idiosyncratic variation across individuals. To mitigate this concern, we had counterbalanced the stories across subjects.

By counterbalancing, the stories for different subjects similarly sampled the semantic space (Supplementary Fig. 1), the semantic categories or relations of interest (Supplementary Figs. 3 and 4), as well as a common set of frequently used words (Supplementary Fig. 2). In addition, the use of audio stories as naturalistic stimuli gave rise to highly reproducible cortical responses across subjects, as shown in prior studies[20] and reinforced by our results (Supplementary Fig. 11). See Supplementary Method 2 for more details on testing the effects of individual variance.

It might appear counterintuitive that some intuitive semantic relations, e.g., similar and contrast, did not map onto any informative voxels despite an adequate sample size (Supplementary Fig. 9). In fact, it is not surprising at all because such relations are both symmetric. For example, (hot, cold) holds a contrast relation, while (cold, hot) also holds the same relation. Likewise, the similar relation is also symmetric. In contrast, other relations, e.g., whole-part, and case relations, are asymmetric. The relation is directed such that flipping two words in a pair changes the relation. Since we use differential vectors to evaluate word relations, our method is more suited for addressing asymmetric relations, instead of symmetric relations.

In this study, the sample size varied across categories or relations. A potential concern might be that the varying sample size could influence the area to which a category or relation was projected. However, this was not a flaw in study design and did not invalidate our findings. Note that the sample-size difference is intrinsic to how English words are distributed across categories or relations. It was our intention to limit our samples to established datasets from published studies with human behavioral data available and associated with words or word relations[29,31]. Moreover, there was no significant correlation between the sample size and the number of voxels that represented a category ($r = 0.0017$, $p = 0.99$) or relation ($r = -0.24$; $p = 0.50$).

Central to this study, we bridge linguistic models and fMRI data during naturalistic audio-story stimuli. Our findings about cortical representations of semantic categories or relations are based on generalizing a predictive model beyond the data used to train the model. While the generalization is supported by our results on model cross-validation and testing, it is desirable to validate some of our model-predicted findings with experimental data in future studies. Importantly, our computational model-based strategy enables high-throughput investigation of how the brain encodes concepts and relations beyond what is feasible for a single experiment. Hypotheses informed by the model may also lend inspiration to future experimental studies. Moreover, it will also be useful to incorporate neurobiological principles and refines the word2vec model in order to improve the correspondence between the word embedding and the word representation on the human cortex.

## Methods

**Subjects, stimuli and experiments.** Nineteen human subjects (11 females, age 24.4 ± 4.8, all right-handed) participated in this study. All subjects provided informed written consent according to a research protocol approved by the Institutional Review Board at Purdue University. While being scanned for fMRI, each subject was listening to several audio stories collected from The Moth Radio Hour (https://themoth.org/radio-hour) and presented through binaural MR-compatible headphones (Silent Scan Audio Systems, Avotec, Stuart, FL). A single story was presented in each fMRI session (6 m 48 s ± 1 min 58 s). For each story, two repeated sessions were performed for the same subject.

Different audio stories were used for training vs. testing the encoding model. For training, individual subjects listened to different sets of stories. When combined across subjects, the stories used for training amounted to a total of 5 h 33 m (repeated twice). This design provided a large number of stimulus–response samples beneficial for training the encoding model, which aimed to map hundreds of semantic features to thousands of cortical voxels. For testing, every subject listened to the same single story for 6 m 53 s; this story was different from those used for training.

In an attempt to sample a sufficiently large number of words in the semantic space, we intentionally chose audio stories of diverse contents. Since different subjects listened to distinct (training) stories, we further counterbalanced the stories across subjects. For different subjects, the stories included different words ($2492 \pm 423$) but sampled similar distributions in the semantic space (Supplementary Fig. 1)[54]. For each semantic category or relation of interest, the associated words were roughly evenly sampled across subjects (Supplementary Figs. 3 and 4). The stories presented to each subject also included a set of common words used frequently in daily life (Supplementary Fig. 2). In total, the training stories include 5228 unique words. By counterbalancing the stories across subjects, we attempted to avoid any notable sampling bias that could significantly confound the idiosyncratic variation across subjects with the variation of the sampled words across subjects. See Supplementary Method 1 for more details.

**Data acquisition and processing.** $T_1$ and $T_2$-weighted MRI and fMRI data were acquired in a 3T MRI system (Siemens, Magnetom Prisma, Germany) with a 64-channel receive-only phased-array head/neck coil. The fMRI data were acquired with 2 mm isotropic spatial resolution and 0.72 s temporal resolution by using a gradient-recalled echo-planar imaging sequence (multiband = 8, 72 interleaved axial slices, TR = 720 ms, TE = 31 ms, flip angle = 52°, field of view = $21 \times 21$ cm$^2$).

Since our imaging protocol was similar to what was used in the human connectome project (HCP), our MRI and fMRI data were preprocessed by using the minimal preprocessing pipeline established for the HCP (using software packages AFNI, FMRIB Software Library, and FreeSurfer pipeline). After preprocessing, the images from individual subjects were co-registered onto a common cortical surface template (see details in[55]). Then the fMRI data were spatially smoothed by using a gaussian surface smoothing kernel with a 2 mm standard deviation.

For each subject, the voxel-wise fMRI signal was standardized (i.e., zero mean and unitary standard deviation) within each session and was averaged across repeated sessions. Then the fMRI data were concatenated across different sessions and subjects for training the encoding model.

**Modeling and sampling the semantic space.** To represent words as vectors, we used a pretrained word2vec model[21]. Briefly, this model was a shallow neural network trained to predict the neighboring words of every word in the Google News dataset, including about 100 billion words (https://code.google.com/archive/p/word2vec/). After training, the model was able to convert any English word to a vector embedded in a 300-dimensional semantic space (extracted through software package Gensim[56] in python). Note that the basis functions learned with word2vec should not be interpreted individually, but collectively as a space. Arbitrary rotation of the semantic space would end up with an equivalent space, even though it may be spanned by different semantic features. The model was also able to extract the semantic relationship between words by simple vector operations[30]. Individual words were extracted from audio stories using Speechmatics (https://www.speechmatics.com/), and then were converted to vectors through word2vec.

**Voxel-wise encoding model.** We mapped the semantic space, as modeled by word2vec, to the cortex through voxel-wise linear encoding models, as explored in previous studies[1,24,38,39]. For each voxel, we modeled its response to a word as a linear combination of the word features in the semantic space.

$$x_i = a_i + \mathbf{b}_i \mathbf{y} + \varepsilon_i, \tag{1}$$

where $x_i$ is the response at the i-th voxel, $\mathbf{y}$ is the word embedding represented as a 300-dimensional column vector with each element corresponding to one axis (or feature) in the semantic space, $\mathbf{b}_i$ is a row vector of regression coefficients, $a_i$ is the bias term, and $\varepsilon_i$ is the error or noise.

**Training the encoding model.** We used the (word, data) samples from the training stories to estimate the encoding model. As words occurred sequentially in the audio story, each word was given a duration based on when it started and ended in the audio story. A story was represented by a time series of word embedding sampled every 0.1 s. For each feature in the word embedding, its time-series signal was further convolved with a canonical hemodynamic response function (HRF) to account for the temporal delay and smoothing due to neurovascular coupling[57]. The HRF-convolved feature-wise representation was standardized and down-sampled to match the sampling rate of fMRI.

It follows that the response of the i-th voxel at time $t$ was expressed as Eq. (2)

$$x_i(t) = a_i + \mathbf{b}_i \mathbf{y}(t) + \varepsilon_i(t). \tag{2}$$

We estimated the coefficients $(a_i, \mathbf{b}_i)$ given time samples of $(x_i, \mathbf{y})$ by using least-squares estimation with L2-norm regularization. That is, to minimize the following loss function defined separately for each voxel.

$$L_i = \frac{1}{T} \sum_{t=1}^{T} (x_i(t) - a_i - \mathbf{b}_i \mathbf{y}(t))^2 + \lambda_i \parallel \mathbf{b}_i \parallel_2^2, \tag{3}$$

where $T$ is the number of temporal samples, and $\lambda_i$ is the regularization parameter for the i-th voxel.

We applied 10-fold generalized cross-validation[25] in order to determine the regularization parameter. Specifically, the training data were divided evenly into ten subsets, of which nine were used for model estimation and one was used for model validation. The validation was repeated ten times such that each subset was used once for validation. In each time, the correlation between the predicted and measured fMRI responses was calculated and used to evaluate the validation accuracy. The average validation accuracy across all ten times was considered as the cross-validation accuracy. We chose the optimal regularization parameter that yielded the highest cross-validation accuracy. Then we used the optimized regularization parameter and all training data for model estimation, ending up with the finalized model parameters denoted as $(\hat{a}_i, \hat{\mathbf{b}}_i)$.

**Cross validating the encoding model.** We further tested the statistical significance of 10-fold cross-validation for every voxel based on a block-wise permutation test[58]. Specifically, we divided the training data into blocks; each block had a 20-s duration. We kept the HRF-convolved word features intact within each block but randomly shuffled the block sequence for each of 100,000 trials of permutation. Before or after the block-wise shuffling, the word feature time series had the nearly identical magnitude spectrum, whereas the shuffling disrupted any word-response correspondence. For every trial of permutation, we ran the 10-fold cross-validation as aforementioned, resulting in a null distribution that included 100,000 cross-validation accuracies with permuted data. Against this null distribution, we compared the cross-validation without permutation and calculated the one-sided $p$ value while testing the significance with FDR $q<0.05$.

Following this 10-fold cross-validation, the model had been validated against 5228 unique words. Thus, at the voxels of statistical significance, the word-evoked responses were considered to be predictable by the encoding models. Using the voxels of significance, we further created a cortical mask and confined the subsequent analyses to voxels in the created mask.

**Testing the encoding model.** We also tested how well the encoding model could be generalized to a new story never used for model training and further evaluated how different regions varied in their responses to the same input stimuli. For this purpose, the trained encoding model was applied to the testing story, generating a voxel-wise model prediction of the fMRI response to the testing story.

$$\hat{x}_i(t) = \hat{a}_i + \hat{\mathbf{b}}_i \mathbf{y}(t), \tag{4}$$

where $\mathbf{y}(t)$ is the HRF-convolved time series of word embedding extracted from the testing story.

To evaluate the encoding performance, we calculated the correlation between the predicted fMRI response $\hat{x}_i$ and the actually measured fMRI response $x_i$. To evaluate the statistical significance, we used a block-wise permutation test[58] (20-s window size; 100,000 permutations) with FDR $q<0.05$, similar to the analysis for cross-validation.

Since the measured fMRI responses to the testing story were averaged across sessions and subjects, the average responses had a much higher signal to noise ratio, or lower noise ceiling[59], allowing us to visually inspect the encoding performance based on the response time series and to exam the response variation across regions. However, the testing story only included 368 unique words. Where the model succeeded in predicting the voxel response to the testing story was expected to be incomplete, relative to where the model would be able to predict given a larger set of word samples, e.g., as those used for cross-validation.

In addition, we extracted the fMRI responses at ROIs predefined in the Human Brainnetome Atlas, which is a connectivity-based parcellation reported in an independent study[27]. We averaged the measured and model-predicted fMRI responses within each given ROI, and compared them as time series (see Fig. 3c). The corresponding statistics regarding the location, size, and prediction performance of each ROI are listed in Supplementary Table 2.

**Mapping cortical representation of semantic categories.** Using it as a predictive model, we further applied the estimated encoding model to a large vocabulary set including about 40,000 words[29]. At each voxel, we calculated the model-predicted response to every word and estimated the mean and the standard deviation for the response population and normalized the model-predicted response to any word as a $z$ value.

Then we focused on the model prediction given 9849 words from nine categories: tool, human, plant, animal, place, communication, emotion, change, quantity (Supplementary Table 3). See Supplementary Method 3 for more details about collecting samples for semantic categories. Every word had been rated for concreteness, ranging from 1 (most abstract) to 5 (most concrete). For each word, we used word2vec to compute its vector representation, and then used the voxel-wise encoding model to map its cortical representation.

As words were grouped by categories, we sought the common cortical representation shared by those in the same category. For this purpose, we averaged the cortical representation of every word in each category, and thresholded the average representation based on its statistical significance (one-sample $t$-test, FDR $q < 0.01$). We evaluated whether a given category was differentially represented by

the left vs. right hemisphere, by counting for each hemisphere the number of voxels associated with that category. We also evaluated the semantic selectivity of each voxel, i.e., how the voxel was more selective to one category than the others. For a coarse measure of categorical selectivity, we identified, separately for each voxel of significance, a single category that resulted in the strongest voxel response among all nine categories and associated that voxel with the identified category (or by "winners take all").

**Assessing word relations in the semantic space**. Vector representations of words obtained by word2vec allow word relations to be readily extracted and applied with simple vector arithmetic[30]. For example, an arithmetic expression of "hand − finger + second" in the semantic space leads to a vector close to that of "hour" in terms of cosine similarity. In this example, the subtraction extracts the relationship between hand and finger, which is intuitively interpretable as a whole-part relationship as a finger is part of a hand. It follows that the addition transfers this whole-part relationship to another word second, ending up with the word hour, while a second is indeed part of an hour.

Beyond this illustrative example, we examined a number of word pairs that held one out of ten classes of semantic relations, as defined in or derived from the SemEval-2012 Task 2 dataset[31]. In this dataset, individual word-pair samples were scored by humans (by crowdsourcing) in terms of the degree to which a word pair could be viewed as an illustrative example of a specific semantic relation. The score ranged from −100 to 100 with −100 being the least and 100 being the most illustrative. For each class of semantic relation, we only included those word pairs with positive scores such that the included word pairs were affirmative samples that matched human understandings in a population level. We excluded the reference relation and separated the space-time relation into space-associated and time-associated relations. In brief, the ten semantic relations (and their sample sizes) were whole-part (178 pairs), class-inclusion (113), object-attribute (63), case relations (106), space-associated (58), time-associated (44), similar (160), contrast (162), object-nonattribute (69), cause-effect (107). See details in Supplementary Table 4. Also see Supplementary Method 3 for details about collecting samples of semantic relations.

We investigated how generalizable semantic relations could be represented by differential vectors in the semantic space by using a leave-one-out test for each class of semantic relation. Specifically, we used the differential vector between any pair of words as the vector representation of their relation (or the "relation vector"). For a given class of semantic relation, we calculated the cosine similarity between the relation vector of every word pair in the class and the average relation vector of all other word pairs in the same class (or the "matched relational similarity") and compared it against the cosine similarity with the average relation vector in a different class (or the "unmatched relational similarity"). The matched relational similarity indicated the consistency of relation vectors in the same class of semantic relation. Its contrast against the unmatched relational similarity was evaluated with paired $t$-test (FDR $q<0.001$). See more details in Supplementary Method 4 and the related results in Supplementary Fig. 6.

**Mapping cortical representation of semantic relation**. Applying the encoding model to the differential vector of a word pair could effectively generate the cortical representation of the corresponding word relation. With this notion, we used the encoding model to predict the cortical representations of semantic relations. For each class of semantic relation, we calculated the relation vector of every word pair in that class, projected the relation vector onto the cortex using the encoding model, and averaged the projected patterns across word-pair samples in the class. For the averaged cortical projection, we tested the statistical significance for every voxel based on a paired permutation test. In this test, we flipped every word pair at random for 100,000 trials. For every trial, we calculated the model-projected cortical pattern averaged across the randomly flipped word pairs, yielding a null distribution per voxel. Against this voxel-wised null distribution, we compared the average voxel value projected from non-flipped word pairs and calculated the two-sided $p$ value with the significance level at FDR $q < 0.05$. The resulting pattern of significant voxels was expected to report the primary cortical representation of each semantic relation of interest.

Complementary to the voxel-wise univariate analysis, we also applied an MVPA to the cortical projection of word relations[60]. Unlike the univariate analysis, MVPA was able to account for interactions between voxels and uncover likely multiple cortical patterns associated with each semantic relation of interest. Specifically, given a class of semantic relation, we concatenated the cortical pattern projected from every word-pair samples in that class and calculated a covariance matrix describing the similarity of representations between samples[61,62]. By using principal component analysis (PCA), we obtained a set of orthogonal components (i.e., eigenvectors), each representing a cortical pattern that accounted for the covariance to a decreasing extent. We chose the top-10 principal components and calculated the pattern-wise cosine similarity between every component and the cortical projection of every word-pair sample. For each component, we averaged the cosine similarity across all samples of the given semantic relation and tested the statistical significance based on one-sample $t$-test ($p<0.01$). Specifically, for any relation with multiple significant components, we grouped and sorted the word pairs based on their corresponding cosine similarities with each component. For each component, we

listed the top-4 word pairs with the highest cosine similarity in order to gain intuitive understanding as to whether the component was selective to a sub-class of that relation. See more details in Supplementary Method 7.

**Reporting summary**. Further information on research design is available in the Nature Research Reporting Summary linked to this article.

## Data availability
The raw and processed imaging datasets, as well as the supplementary data that support the findings of this study, are shared via a public repository in the Open Science Framework (https://osf.io/eq2ba/). The DOI of this dataset is https://doi.org/10.17605/OSF.IO/EQ2BA. The raw imaging datasets will also be shared via the OpenNeuro platform (https://openneuro.org/).

## Code availability
The code for training and testing the voxel-wise encoding model is also shared via the public repository in the Open Science Framework (https://osf.io/eq2ba/).

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

## Acknowledgements
The authors thank David Kemmerer for helpful discussions and comments to this paper. This work was supported by National Institute of Mental Health R01MH104402, Purdue University, and the University of Michigan.

## Author contributions
Z.L., Y.Z., and R.W. designed research; Y.Z. and K.H. performed research; Y.Z. analyzed data; Y.Z. and Z.L. wrote the paper.

## Competing interests
The authors declare no competing interests.
