## [Peer Review File · Nature Communications]

Reviewers' Comments:

Reviewer #1:

Remarks to the Author:

In this paper, the authors use natural language encoding models for fMRI to examine representations of specific semantic relations, such as "whole-part". Overall, the methods used here are solid and the analyses are clever. However, I would like to see more validation of the semantic relation analyses before recommending acceptance.

For the semantic relation analysis I have three major questions:

(1) does the semantic relation/analogy vectors generalize in the embedding space? e.g. if you build the part-whole difference vector using most of the word pairs, can you use it to predict parts from the whole on other words? It would be nice to see both qualitative examples of this (a few examples) and quantitative evaluation (e.g. the average rank of the part word inferred from the whole using the whole-part vector, perhaps compared to the average rank of the same word using a different semantic relation vector).

(2) to what degree is this contrast driven by a few words? The t-test used to assess significance on this analysis is not robust to outliers. It would be much better to do a randomization test such as permutation, where, e.g. the wholes vs. parts are randomly flipped and used to generate new vectors.

(3) I'm concerned that the semantic relation vectors are (maybe strongly) correlated with specific categories. For example, the "space-associated" map seems to be very similar to the "place" map shown in Figure 2. This particular example is unsurprising, but it raises the question of whether the semantic relations in general are specific to certain categories. It would be nice to show which categories each semantic relation vector is associated with, even among the 9 categories used in the earlier analysis.

Reviewer #2:

Remarks to the Author:

This paper uses an encoding model based on word2vec features to predict voxel responses to words presented in fMRI. The trained encoding model is then fed new stimuli which are projected onto the fMRI voxel space. I generally appreciate the data-driven approach taken here, and the encoding prediction accuracy using the word2vec features is certainly impressive, but my concern is that the encoding model is then over-generalised to regions where it could not predict responses in the first place.

My main comment is that the interpretation of the model projections is limited by the model fit. That is, if the encoding model does a poor job of predicting voxel responses, then how much confidence should be placed in its weights, and following projections? This is especially a concern when projecting new words and semantic relationships through these weights. At the extremes, as the encoding model is only predicting responses of a subset of voxels above chance (i.e. Fig 1a), then interpreting the model weights for voxels outside this subset is not very informative (e.g., Fig 2 shows more areas than Fig 1a) because the model cannot make accurate predictions for those areas.

Throughout the manuscript, the authors refer to significant weight projection as an area's representation, or the representation of a particular concept in an area. For example: L122: "we also mapped the cortical representations of other semantic relationships" L153: "semantic relations were represented by anti-correlated cortical networks" This wording is misleading, and should be rephrased

to reflect the correlational nature of the analysis. In addition, this limitation on the interpretation of these results should be included in the discussion.

How does the number of samples used for the projection affect the results? Different category projections are based on different numbers of words (e.g., 938 versus 5138 is a big difference). The same holds for the different semantic relationships.

L340. The correlations resulting from the 20-fold cross-validation are not independent because they use overlapping training data, so a permutation test would be more appropriate than a t-test

L398. Unclear why a paired t-test was used here instead.

L528: Data availability, the authors write "The imaging datasets and processing codes will be publicly available later." – This is a bit vague, and I believe a more concrete statement is warranted (e.g., specify the number of months after publication). Also, at least the code should be made available for review.

Reviewer #3:

Remarks to the Author:

Review of "Connecting concepts in the brain: Mapping cortical representations of semantic relations" for Nature Communications (double-blind peer-review)

SUMMARY

This paper describes an fMRI study aimed at mapping conceptual relationships (e.g., "whole-part") onto cortical networks. Nineteen participants each listened to audio stories from The Moth Radio Hour, including several stories that were unique to each participant (for training an encoding model) along with one story that was presented to all participants (for testing the model). A semantic space was based on a pre-trained word2vec model of word embeddings that represents words as 300-dimension vectors of semantic features, and a voxel-wise encoding model related this semantic space to the fMRI data by expressing each voxel's response as a weighted sum of semantic features. This resulted in whole-brain maps of regression coefficients corresponding to each concept. Within these voxelwise maps, the left hemisphere appeared to be generally more selective for concrete concepts, while the right hemisphere appeared to be more selective for abstract concepts. From there, the authors operationalized conceptual relationships as the average of voxel-wise subtractions between whole-brain maps corresponding to pairs of individual words. While a few intuitive conceptual relationships surprisingly did not correspond to voxelwise differences anywhere in the brain, most contrasts revealed significantly positive or negative to z-statistics widely distributed throughout the brain. The authors interpret these differences as anti-correlated cortical networks and observe that several of the contrasts appear to distinguish between areas generally associated with either the default mode network or the frontoparietal attention network.

This is a very interesting and creative marriage of computational and linguistic models with fMRI data that has the potential to provide new and important insights into neural representation. However, I have several concerns regarding the analysis approach and the interpretation of findings that limit my enthusiasm for the manuscript in its current form.

CONCERNS

1) Content and participant are confounded in training data.

To train the voxelwise encoding model, data was concatenated across participants who each listened to different sets of stories about diverse topics. Specifically, each of the 19 participants contributed about 18 minutes of unique stories to the combined 5 hours 33 minutes of training data. For testing, every subject listened to the same 7-minute story.

I appreciate that by having participants listen to different stories and then combining data across participants, the authors were able to more thoroughly sample many more unique words than they would have been able to if each participant listened to the same set of audio stories. However, even though the model was always tested based on data from the same story, I am concerned that individual differences (e.g., in laterality, which can be quite variable across participants) are confounded with the words used to train the model.

For instance, Table S2 displays the number of paired samples for each semantic relationship (ranging from 44 to 178). Ideally, each semantic relationship was evenly sampled across participants. Otherwise, if a subset of participants is overrepresented for one semantic dimension, and a different subset of participants is overrepresented for a different semantic dimension, then the trained encoding model will reflect the individual differences of the participants.

Notably, this is a different approach than in previous studies that the authors cite such as Huth et al. (2016, Nature). In that study, each participant listened to the same 10 audio stories for model estimation. Along with providing an unbiased training set, exposing participants to the same stimuli allowed previous authors to separate variance that was consistent across participants from the idiosyncratic variance of each participant (e.g., as displayed in Extended Data Figure 2).

With enough data, the appropriate counter balancing and the necessary measure checks, confounds from training the model in this way could potentially wash out. However, there is currently no discussion of how the authors controlled for this. The authors are not specific about how they selected the stimuli, other than noting that the audio stories were intentionally of diverse topics.

2) Intuitive semantic relations (e.g., "similar") do not appear to map onto any informative voxels despite adequate power.

The authors note that "not all human-defined semantic relations have an underlying vector field that can be represented as a consistent cortical region or network" (p. 12). Indeed, in Table S2, the semantic relations "similar" and "contrast" each map onto 0 voxels, and the relation "cause-effect" maps onto just 11 voxels (compared to >12,000 voxels for several of the other semantic relations like "class-inclusion"). This is puzzling to me, especially since the dataset includes more word pairs for "similar" and "contrast" than for most other relations. It would be helpful for the authors to determine (or at least more specifically speculate) why not all relations from the SemEval-2012 dataset map onto cortical networks.

I can imagine a post-hoc explanation for why there might be 0 reliable voxels for "similar" (e.g., "house" vs. "home"). For instance, the cortical representations of "house" and "home" could be so nearly identical that differences between these brain states are negligible. However, theoretically, I have trouble imagining why there would also be zero significant voxels for "contrast" (e.g., "hot" vs. "cold"). If the authors could convincingly explain why some intuitive conceptual relationships are distinctly represented whereas other intuitive relationships are not represented at all, this would be, in my opinion, the most interesting and important finding of the paper. I am concerned, however, the dissociations reflect deficiencies and confounds in the model (e.g., as described in my first concern) as

opposed to new discoveries about the brain.

3) Exclusively voxelwise approach to examine distributed representations.

Ideas throughout the paper are focused on distributed neural representations. However, the encoding model used to examine distributed neural representations is univariate (i.e. each voxel is separately modeled), with hypotheses tested by counting or visualizing the number of individual voxels labeled by each category or relation. Although terms such as “anticorrelated networks” suggest analysis of how different parts of the brain interact in different contexts, analyses are limited to the selectivity of individual voxels visualized as schematic maps of the brain and interpreted in terms hemispheric asymmetries and the relative loading of feature dimensions.

Critically, using this univariate approach, the authors were not able to identify voxelwise differences for seemingly basic and intuitive relationships like similarity and contrast. If the model is correct (but see my first and second concerns above), it could be that these relationships are represented by patterns and interactions across voxels and regions as opposed to the individual tuning properties of any one voxel or region. In this case, widely used multivariate approaches to fMRI analysis like multivariate pattern analysis and functional connectivity may be important to consider as well (and may also be more consistent with ideas of representation). For instance, representational similarity analysis (e.g., Kriegeskorte et al., 2008 *Frontiers*) could also be applied to further compare between word2vec representations and brain representations for intuitive and yet seemingly intractable semantic relationships like similarity and contrast.

4) ROI model fits displayed in Figure 1 appear to be inconsistent or circular with whole-brain statistics.

To show that the word2vec-based encoding model was able to capture the quantitative and generalizable relationships between cortical responses and word attributes the stimulus set, the authors display overlays of the BOLD timeseries and model predictions for various regions (Fig 1b). However, these overlays appear to depict much higher correlations than suggested by the whole-brain statistics displayed in Fig 1a. For instance, the first pair of lines representing IFG suggests to me an exceptionally strong correlation between the signal and model prediction. Yet, the color scale for Fig 1a ranges from 0.3 to 0.6, with most of both left and right IFG completely masked out (i.e., a correlation < 0.3). That is, right IFG is almost entirely empty and left IFG also appears to be mostly empty with the exception of a very anterior cluster bordering IFG. This suggests that the signal timeseries (and its relationship to the model prediction) could not possibly reflect IFG as it would be anatomically defined. Possibly, the signal timeseries specifically reflects the significant cluster of voxels that are within proximity to left IFG. If so, “double dipping” gives the impression that the model is much more accurate than it actually is.

While I’m sensitive to circular inference in neuroimaging, this might otherwise be a minor concern because it can be easily addressed and the authors do not actually report the corresponding statistics. However, because the model fits are so prominently displayed in the first figure and provided as the primary evidence that the model is highly accurate, they strongly influence the reader’s interpretation of subsequent results.

We would like to thank the reviewers for their efforts and comments. We find the review comments
constructive and helpful. Through 3 months of work, we have substantially revised the manuscript in
order to address every concern raised by each reviewer.

The revised version has included additional results as presented in

- • 4 new or revised figures in the main text
- • 11 new figures and 4 new tables in the supplementary information

We believe that our manuscript has been greatly improved after this very substantial revision. The
followings are our point-by-point replies to the review comments. The original comments are shown in
black; our replies are in blue. Our revision to the paper is highlighted in red. In our reply, we also refer to
the revised text, figure, or table.

Reviewers' comments:

Reviewer #1 (Remarks to the Author):

In this paper, the authors use natural language encoding models for fMRI to examine representations of
specific semantic relations, such as “whole-part”. Overall, the methods used here are solid and the
analyses are clever. However, I would like to see more validation of the semantic relation analyses
before recommending acceptance.

Thank you. We are delighted that the reviewer is overall positive. We are grateful not only for the
reviewer’s questions but also for the constructive suggestions that guide us to address those questions.
Given the review comments, we have done “more validation of the semantic relation analyses”, as
elaborated in our responses to the following three specific questions.

For the semantic relation analysis I have three major questions:

(1) does the semantic relation/analogy vectors generalize in the embedding space? e.g. if you build the
part-whole difference vector using most of the word pairs, can you use it to predict parts from the
whole on other words? It would be nice to see both qualitative examples of this (a few examples) and
quantitative evaluation (e.g. the average rank of the part word inferred from the whole using the whole-
part vector, perhaps compared to the average rank of the same word using a different semantic relation
vector).

Thank you for the excellent questions. To answer these questions, we have done additional analyses on
the semantic relation vectors in the embedding space, as suggested by the reviewer. The new results are
consistent with and supportive of our original methods, results, and conclusions.

We analyzed the generalizability of semantic relation vectors in the embedding space. As suggested by
the reviewer, we performed a “leave-one-out” test of how word relation vectors could be generalized
across word pairs of the same semantic relation (“matched”) vs. those of different semantic relations
(“unmatched”).

For example, <book, pages> is a word-pair that holds a “whole-part” relation, and their relation vector is
the vector difference of the two words in the embedding space. We asked whether this relation vector
was similar to the relation vector averaged across other word-pairs in the same “whole-part” relation,

while leaving out one pair, e.g. <book, pages>. We used cosine similarity to measure this relational
similarity, ranging from -1 to 1. In this example, the relational similarity between <book, pages> and
other word pairs in the same semantic relation was 0.36.

We repeated this measure by using each word-pair as the leave-one-out test example and averaged the
resulting similarity across different choices of the test example. This average similarity was referred to as
the “matched relational similarity”.

As suggested by the reviewer, we compared the “matched relational similarity” against the “unmatched
relational similarity”, which was based on the relational similarity between two word-pairs of different
semantic relations. For example, we measured the relational similarity between <book, pages> and
<fire, hot>, which were unmatched since the former held a “whole-part” relation and the latter held an
“object-attribute” relation.

We compared the matched vs. unmatched relational similarity and evaluated their statistical difference
with the paired t-test. In the revised paper, Supplementary Fig. 6 shows the matched relational similarity
in contrast to the unmatched relational similarity. In general, the matched relational similarity was
significantly higher than the unmatched relational similarity for 6/10 semantic relations: “whole-part”,
“class-inclusion”, “object-attribute”, “case relations”, “space-association”, and “time-association”.

This result suggests that each of these 6 semantic relations has a vector representation in the
embedding space that are reasonably generalizable across its examples. In contrast, we did not find a
similarly generalizable relation vector for 4/10 semantic relations: “similar”, “contrast”, “object-non-
attribute”, and “cause-effect”. This might in part explain why we could not find any informative cortical
patterns associated with those relations.

In addition, we also asked how well we could classify the semantic relation between a given pair of
words based on their vector difference in the embedding space. To address this classification problem,
we calculated the vector representation of each semantic relation by averaging the relation vectors of
all word pairs that hold that semantic relation; then, we calculated the cosine similarity between the
relation vector of the given word pair and the vector of every semantic relation; we identified the (top-
1) semantic relation that was most similar to the given word relation. Note that for any given word-pair,
we excluded it from the examples of its corresponding semantic relation to avoid any minor bias. We
ran this classification task for every word-pair considered in this study and evaluated the classification
accuracy separately for each semantic relation, as shown in Supplementary Fig. 7.

The top-1 classification accuracy was found to be higher than 75% for 4/10 semantic relations: “class-
inclusion”, “case relations”, “space-associated” and “time-associated”. The classification accuracy for
“whole-part” or “object-attribute” was relatively lower: 40% or 59%, respectively; however, it was still
much larger than the chance level (i.e. 10%). By closer inspection, we found that the mis-classified word
pairs tended to hold more than one relation. For example, <bedroom, bed>, <library, books>, <zoo,
animal>, <bank, money>, <garden, vegetables> were included as example word pairs for the whole-part
relation; apparently these word pairs were also space-associated and indeed were similar to the “space-
association” relation in terms of the cosine similarity. Similarly, some word pairs, such as <athlete, fit>,
<millionaire, riches>, <pacifist, peace>, hold both “object-attribute” and “cause-effect” relations.

Taken together, the results on the matched vs. unmatched relational similarity (Supplementary Fig. 6)
and the top-1 classification accuracy of semantic relation (Supplementary Fig. 7) suggest that the
“semantic relation vectors generalize in the embedding space”, for some semantic relations, but not for
others. The generalizable relations include but are not limited to “whole-part”, “class-inclusion”,
“object-attribute”, “case relations”, “space-association”, and “time-association”. We believe the new
analyses and results have addressed the reviewer’s questions and have further lent support to cortical
mapping of a semantic relation by linear projection of its vector representation in the word2vec
embedding space.

Accordingly, we have made the following revision to the main text.

From line 136 to 144 in Results

*“We chose word-pairs from the SemEval-2012 Task 2 dataset [31]. Every chosen word-pair had been*
*human rated as an affirmative example of one of 10 classes of semantic relation: “whole-part”, “class-*
*inclusion”, “object-attribute”, “case relations”, “space-associated”, “time-associated”, “similar”,*
*“contrast”, “object-nonattribute”, and “cause-effect” (Supplementary Table 4). For the first 6 classes,*
*the relation vectors in the semantic space were found to be more consistent across word-pairs in the*
*same class than those in different classes (Supplementary Fig. 6). For each of the first 6 classes, the*
*relation between every pair of words could be identified based on the relation vectors of the other word-*
*pairs in the same class, rather than those from any different class, with the top-1 accuracy from 40 to*
*83% against the 10% chance level (Supplementary Fig. 7).”*

From line 276 to 277 in Discussion

*“Our results suggest that the direction of relation vector tends to be generalizable and*
*transferrable across word pairs in the same semantic relation (Supplementary Fig. 6 & 7).”*

From line 528 to 534 in Methods

*Vector representations of words obtained by word2vec allow word relations to be readily extracted and*
*applied with simple vector arithmetic [30]. For example, an arithmetic expression of (“hand” - “finger” +*
*“second” leads to a vector close to that of “hour” in terms of cosine similarity. In this example, the*
*subtraction extracts the relationship between “hand” and “finger”, which is intuitively interpretable as a*
*“whole-part” relationship as a “finger” is part of a “hand”. It follows that the addition transfers this*
*“whole-part” relationship to another word “second”, ending up with the word “hour”, while a “second” is*
*indeed part of an “hour”.*

From line 548 to 557 in Methods

*“We investigated how generalizable semantic relations could be represented by differential vectors in*
*the semantic space by using a leave-one-out test for each class of semantic relation. Specifically, we used*
*the differential vector between any pair of words as the vector representation of their relation (or the*
*“relation vector”). For a given class of semantic relation, we calculated the cosine similarity between the*
*relation vector of every word-pair in the class and the average relation vector of all other word pairs in*
*the same class (or the “matched relational similarity”) and compared it against the cosine similarity with*
*the average relation vector in a different class (or the “unmatched relational similarity”). The matched*
*relational similarity indicated the consistency of relation vectors in the same class of semantic relation.*

*Its contrast against the unmatched relational similarity was evaluated with paired t-test (FDR $q < 0.01$).*
*See more details in Supplementary Method 4 and the related results in Supplementary Fig. 6.”*

(2) to what degree is this contrast driven by a few words? The t-test used to assess significance on this
analysis is not robust to outliers. It would be much better to do a randomization test such as
permutation, where, e.g. the wholes vs. parts are randomly flipped and used to generate new vectors.

Thank you for this excellent question and the constructive suggestion to address this question. As
suggested by the reviewer, we performed a permutation test for non-parametric evaluation of the
statistical significance when mapping the cortical representation of each semantic relation.

For each time of permutation, we randomly flipped the direction of every word pair. For example, the
relation vector of *<book, pages>* could be the element-wise subtraction between word vectors of “book”
and “pages” or between word vectors of “pages” and *book*” with equal probability. For a given semantic
relation of interest, we mapped the randomly flipped relation vector of every example word-pair and
averaged the resulting cortical representation across word-pairs. By repeating this process for
$N=100,000$ times, we generated a null distribution for representation at every voxel. Against this null
distribution, we compared the cortical representation obtained with intact word-pairs (without any
flipping). We calculated the probability that the samples in the null distribution exceeded the cortical
representation of interest and chose the significant voxels with FDR $q < 0.05$.

Based on this non-parametric statistic test, we have updated our Figure 6 & 7 and added Supplementary
Figure 9. For each semantic relation, the number of significant voxels is listed in Supplementary Table 4.
In general, the new results obtained with the permutation test are consistent with our previous results
using the paired t-test. These results suggest that the relational contrast is not driven by a few words.

Accordingly, we have made the following revision to the main text.

From line 151 to 154 in Results

*“Nevertheless, their pairwise relations all entailed the “whole-part” relation, as illustrated in Fig. 6a. By*
*using the encoding model, we mapped the pairwise word relationship onto voxels in the semantic system*
*(as shown in Fig. 2), averaged the results across pairs, and highlighted the significant voxels (paired*
*permutation test, FDR $q < 0.05$).”*

From line 565 to 571 in Methods

*“For the averaged cortical projection, we tested the statistical significance for every voxel based on a*
*paired permutation test. In this test, we flipped every word pair at random for 100,000 trials. For every*
*trial, we calculated the model-projected cortical pattern averaged across the randomly flipped word*
*pairs, yielding a null distribution per voxel. Against this voxel-wised null distribution, we compared the*
*average voxel value projected from non-flipped word pairs and calculated the two-sided p-value with the*
*significance level at FDR $q < 0.05$. The resulting pattern of significant voxels was expected to report the*
*primary cortical representation of each semantic relation of interest.”*

(3) I'm concerned that the semantic relation vectors are (maybe strongly) correlated with specific
categories. For example, the “space-associated” map seems to be very similar to the “place” map shown
in Figure 2. This particular example is unsurprising, but it raises the question of whether the semantic

relations in general are specific to certain categories. It would be nice to show which categories each
semantic relation vector is associated with, even among the 9 categories used in the earlier analysis.

Thank you for this comment. Indeed, some semantic relations, such as “space-associated” and “time-
associated”, are strongly correlated with specific categories, like “place” and “quantity”. We agree with
the reviewer that in these two examples, the association between relation and category is natural (or
unsurprising).

However, semantic relations are not always specific to or associated with certain categories. In the
embedding space, the representation of a semantic relation is obtained by taking the difference
between the representations of two concepts, while removing the common representation shared by
two concepts. If the two concepts are in the same semantic category, this differential representation
discounts their shared representation that corresponds to the category.

As the reviewer suggested, we have added **Supplementary Figure 10** showing the percentage by which
the word-pair samples of each semantic relation are associated with each semantic category (among the
9 categories used in this study). For each semantic relation, we counted the number of word-pair
samples associated with every semantic category and converted the number into a percentage value.
See **Supplementary Method 5**.

As shown in Supplementary Fig. 10, there was no or little association with any semantic category for
most semantic relations, e.g. whole-part, class-inclusion, object-attribute. The exceptions were “space-
associated” and “time-associated”, which were associated with the “place” and “quantity”, respectively.

Accordingly, we have made the following revision to the main text.

From line 299 to 304 in Discussion

*“Our results indicate that the “space-associated” relation and the “place” category are represented by*
*similar cortical patterns (Fig. 4f & Fig. 7e). This is unsurprising because the “space-associated” word pairs*
*are often associated with “place”. Such a relation-category association is intrinsic to natural language*
*statistics, and similarly applies to “time-associated” and “quantity”. However, the semantic relations are*
*not in general associated with specific semantic categories (Supplementary Fig. 10). See Supplementary*
*Method 5 for more details about evaluating the association between relations and categories.”*

Reviewer #2 (Remarks to the Author):

This paper uses an encoding model based on word2vec features to predict voxel responses to words
presented in fMRI. The trained encoding model is then fed new stimuli which are projected onto the
fMRI voxel space. I generally appreciate the data-driven approach taken here, and the encoding
prediction accuracy using the word2vec features is certainly impressive, but my concern is that the
encoding model is then over-generalised to regions where it could not predict responses in the first
place.

Thank you for your summary, positive evaluation, and appreciation of our data-driven approach. We
understand and agree with the reviewer’s concern that the encoding model should not be generalized
to regions where it could not predict responses in the first place. This is why we have acquired a large

amount of data to train the model and maximize its ability to predict responses at as many regions as
possible, covering the entire semantic system. We are also cautious about the statistical analysis in
attempts to avoid over-generalization. In this revision, we have refined our analyses according to the
reviewer's comments and suggestions, as elaborated below.

My main comment is that the interpretation of the model projections is limited by the model fit. That is,
if the encoding model does a poor job of predicting voxel responses, then how much confidence should
be placed in its weights, and following projections? This is especially a concern when projecting new
words and semantic relationships through these weights. At the extremes, as the encoding model is only
predicting responses of a subset of voxels above chance (i.e. Fig 1a), then interpreting the model
weights for voxels outside this subset is not very informative (e.g., Fig 2 shows more areas than Fig 1a)
because the model cannot make accurate predictions for those areas.

Thank you for the comments. We understand the concern and would like to address it by 1) clarifying
our results about the model fit and 2) refining our statistical analysis as suggested by the reviewer.

We evaluated the model fit based on testing and cross-validation for distinct purposes. Since cross-
validation utilized many more words than testing, we used voxels with significant cross-validation
accuracy to define a mask of the semantic system and confined subsequent model prediction to voxels
in the mask. This mask was broader than the regions with significant testing accuracy. See below for
more elaboration.

For testing, we trained the encoding model with our training data and tested it against new testing data
never used for model training. This strategy ensured an unbiased evaluation of the model. However, the
testing data were associated with a single story, which only included 368 unique words and sampled a
subspace of the semantic space. As such, the cortical regions where the model was successful in
predicting the fMRI responses to the testing story were expected to cover a part of the brain's semantic
system, whereas the entirety of the semantic system should encode all words or the whole semantic
space. The regions highlighted in **Figure 3a** (or Fig. 1a in the original manuscript) were not completely
inclusive. The model-predictable regions should include but not be limited to those in **Figure 3a**.

By our study design, we did not intend to make the testing results as spatially inclusive as possible;
instead, we focused on evaluating the testing results in time. Since the fMRI signal is generally known to
be noisy, we tried to push the "noise ceiling" by using the same testing story for all subjects and
averaging the resulting voxel time series across subjects. By doing so, we wanted the so averaged voxel
time series to faithfully reflect the story-elicited responses, while suppressing unwanted variation or
noise, since the model should at best be able to predict the story-elicited response instead of the noise.
As shown in **Figure 3c**, the measured and model-predicted response time series demonstrate 1)
different brain regions exhibited highly distinctive responses to the same story, and 2) the encoding
model was able to predict the ROI-level response despite the response variation across regions.

As the reviewer implied, we indeed wanted to find out a more inclusive map of brain regions in which
the model was able to predict the responses to natural language. For this purpose, we focused on cross
validating the encoding model by holding out a varying part of the training data for (10-fold) cross-
validation. This strategy allowed us to test the model performance with many more stories (51) and
words (>5,200) than were available in the testing story, allowing for more comprehensive sampling of
the semantic space. As a result, we were able to identify more brain regions that encoded the semantic

space, indeed as demonstrated in Figure 2. From the result obtained with 10-fold cross validation, we
created a cortical mask - highlighting what we considered as the brain's semantic system. When we
generalized the encoding model to new words or word relations (Figure 4, 5, 6, 7, 8, and Supplementary
Fig. 9), we confined our analysis to voxels within the semantic system as covered by the mask to avoid
over-generalization.

In addition to the above explanation, we refined our method for testing the statistical significance of
cross validation by using a block-wise permutation test (Adolf et al., 2014), according to another
suggestion raised by the reviewer. See more details in our reply to a subsequent comment.

Accordingly, we have made the following revision to the main text.

From line 78 to 84 in Results

*“By 10-fold cross-validation, the model-predicted response was significantly correlated with the*
*measured fMRI response (block-wise permutation test, false discovery rate or FDR $q < 0.05$) for voxels*
*broadly distributed on the cortex (Fig. 2). The voxels highlighted in Fig. 2 were used to delineate an*
*inclusive map of the brain's semantic system, because the cross-validation was applied to a large set of*
*(5,228) words, including those most frequently used in daily life (Supplementary Fig. 2). This map of*
*semantic system, hereafter referred to as the semantic system, was widespread across regions from both*
*hemispheres, ...”*

From line 104 to 105 in Results:

*“We confined the model prediction to the voxels in the semantic system for which the model fit was*
*significant during cross-validation (Fig. 2).”*

From line 152 to 153 in Results:

*“By using the encoding model, we mapped the pairwise word relationship onto voxels in the semantic*
*system (as shown in Fig. 2),”*

From line 478 to 481 in Methods:

*“Following this 10-fold cross-validation, the model had been validated against 5,228 unique words. Thus,*
*at the voxels of statistical significance, the word-evoked responses were considered to be predictable by*
*the encoding models. With the voxels of significance, we further created a cortical mask and confined the*
*subsequent analyses to voxels in the created mask.”*

From line 494 to 499 in Methods:

*“Since the measured fMRI responses to the testing story were averaged across sessions and subjects, the*
*average responses had a much higher signal to noise ratio, or lower “noise ceiling” [58], allowing us to*
*readily inspect the encoding performance over time and to exam the response variation across regions.*
*However, the testing story only included 368 unique words. Where the model succeeded in predicting the*
*voxel response to the testing story was expected to be incomplete, relative to where the model would be*
*able to predict given a larger set of word samples, e.g. as those used for cross validation.”*

Throughout the manuscript, the authors refer to significant weight projection as an area's
representation, or the representation of a particular concept in an area. For example: L122: "we also
mapped the cortical representations of other semantic relationships" L153: "semantic relations were
represented by anti-correlated cortical networks" This wording is misleading, and should be rephrased
to reflect the correlational nature of the analysis. In addition, this limitation on the interpretation of
these results should be included in the discussion.

Thank you for the comment. We agree the paper could benefit from more clarification in wording to
avoid misleading or confusion.

To clarify, we represent a word as a vector and use the encoding model to project the vector to the
cortex, and therefore we interpret and refer to the projected cortical pattern as the cortical
representation of the word or the cortical response to the word. When we cross validate or test the
cortical representation, we calculate the correlation between the model-predicted and measured
response time series given a sequence of words in a story. Although the validation and testing are based
on correlational analysis and metrics, we focus most of the analysis on the model-predicted responses
or representations themselves.

Given the reviewer's suggestion, we have rephrased our wording by using the terms of "cortical patterns
or projections" to indicate the results obtained with projecting word/category/relation representations
to the cortex through the encoding model, while using the terms of "cortical representations" as our
interpretation of the projected cortical patterns. Similarly, we have rephrased the wording about "anti-
correlated networks" by using "co-occurring activation and deactivation" while using "anti-correlated
networks" only when we discuss about our interpretation of those patterns.

We also admit and discuss the limitation that the "cortical representations" are model-predicted in the
current study. It would be desirable to validate some of the model-predicted representations of
semantic categories or relations with experimental data.

Accordingly, we have made the following revision to the main text.

From line 107 to 109 in Results:

*"This map represented each category being projected from the semantic space to the cortex, and thus*
*was interpreted as the model-predicted cortical representation of the category."*

From line 154 to 156 in Results:

*"The resulting cortical map represented each semantic relation being projected from the semantic space*
*to the cortex, reporting the model-predicted cortical representation of the relation."*

From line 160 to 164 in Results:

*"The co-activation and deactivation pattern indicated that conceptual progression from "part" to*
*"whole" manifested itself as increasing deactivation of FPN alongside increasing activation of DMN,*
*whereas progression from "whole" to "part" was shown as the reverse cortical pattern varying in the*
*opposite direction"*

From line 284 to 293 in Discussion:

*“Such cortical patterns often manifest themselves as co-occurring activation and deactivation of regions*
*(Fig. 6, 7 & 8). We interpret this co-activation and co-deactivation as an emerging pattern when the*
*brain relates two concepts that hold a meaningful relation, reflecting the progression from one concept*
*to the other. This pattern encodes generalizable differential relations between concepts, as opposed to*
*concepts themselves, serving differential coding that transcends any conceptual domain or category (Fig.*
*6). Speculatively, this network-based coding of semantic relation is an important mechanism that*
*supports analogical reasoning [45], e.g. matching similar relations with different word-pair samples [31].*
*This brain mechanism might further facilitate humans learning new concepts by connecting them to*
*existing concepts through established semantic relations.”*

From line 307 to 316 in Discussion:

*“We interpret the co-activation/deactivation patterns as “anti-correlated networks”. This interpretation*
*is reasonable given the notion of “activate together wire together”. Task-related patterns of cortical*
*activation resemble those emerging from spontaneous activity or resting state networks [46]. In the*
*context of semantics, the anti-correlated networks reported herein encode a semantic relation, or the*
*direction in which one concept relates to another. For example, conceptual progression from “part” to*
*“whole” has a cortical signature as co-occurring activation in DMN and deactivation in FPN (Fig. 6b). The*
*opposite direction from “whole” to “part” involves the same regions or networks but reverses their*
*polarity in terms of activation or deactivation. In this example, the cortical co-activation/deactivation*
*pattern is nearly identical to the anti-correlated networks observed with resting state fMRI [47], and*
*therefore it is likely to be intrinsic and supported by underlying structural connections.”*

From line 370 to 377 in Discussion:

*“Central to this study, we bridge linguistic models and fMRI data during naturalistic audio-story stimuli.*
*Our findings about cortical representations of semantic categories or relations are based on generalizing*
*a predictive model beyond the data used to train the model. While the generalization is supported by our*
*results on model cross-validation and testing, it is desirable to validate some of our model-predicted*
*findings with experimental data in future studies. Importantly, our computational model-based strategy*
*enables high-throughput investigation of how the brain encodes concepts and relations beyond what is*
*feasible for a single experiment. Hypotheses informed by the model also lend inspiration to future*
*experimental studies.”*

How does the number of samples used for the projection affect the results? Different category
projections are based on different numbers of words (e.g., 938 versus 5138 is a big difference). The
same holds for the different semantic relationships.

Thank you for this critical comment.

Indeed, different categories include different numbers of words. The unequal sample sizes reflect how
frequently used English words are categorized, as opposed to any bias imposed by us. In this study, we
chose English words from established data sets documented in peer-reviewed papers (Miller, 1998;
Brysbaert et al., 2014). On the basis of these existing datasets, we have already maximized the number
of words for each category. See Supplementary Table 3 for more information about the samples in each
category. See Supplementary Method 3 for more details about how the samples were obtained.

Although the sample size was different across categories, our data suggest that the sample size was
large enough for every category. To support this claim, we tested the effect of the sample size on the
number of voxels onto which each category was projected through the encoding model. Specifically, for
a given category, we identified the voxels, at which the model-predicted responses were significant
(one-sample t-test, $FDR < 0.01$) for a varying number of words included in that category.

For example, the maximum number of words in the “animal” category was 734, of which a subset was
selected with the sample size ranging from 10% to 100% of the original sample size. As shown in
**Supplementary Figure 5**, an increasing sample size tended to yield an increasing number of significant
voxels; however, the trend reached or approached the plateau at 60% to 90% of the original sample
size. Therefore, despite the varying sample size per category, the number of samples was large enough
to map the cortical areas associated with every category investigated in this study.

Further, a category that included more words did not turn out to be projected onto more voxels. As
shown in **Supplementary Table 3**, the “tool” category included 200 words and was projected onto 7,906
voxels, whereas the “change” category included 3,417 words but was only projected onto 5,768 voxels.
Overall, there was no correlation ($r = 0.0017$, $p = 0.9965$) between the number of word samples in a
category and the number of voxels that the category was projected onto. Similarly, there was no
significant correlation between the number of word-pair samples in a relation and the number of voxels
that the relation was projected onto ($r = -0.2426$, $p = 0.4995$). Also see **Supplementary Table 4**.

Accordingly, we have made the following revision to the main text.

From line 116 to 119 in Results:

*“Although the size of word samples varied across categories (Supplementary Table 3), the sample size*
*was sufficiently large for every category, since the resulting category representation had reached or*
*approached its maximum extent at the given sample size (Supplementary Fig. 5).”*

From line 363 to 369 in Discussion:

*“In this study, the sample size varied across categories or relations. A potential concern might be that the*
*varying sample size could influence the area to which a category or relation was projected. However, it is*
*worth noting that sample-size difference is intrinsic to how English words are distributed across*
*categories or relations. We intentionally limited our samples to established datasets from published*
*studies with human behavioral data available and associated with words or word relations [29, 31].*
*Moreover, there was no significant correlation between the sample size and the number of voxels that*
*represented a category ($r=0.0017$, $p=0.99$) or relation ($r=-0.24$; $p=0.50$).”*

L340. The correlations resulting from the 20-fold cross-validation are not independent because they use
overlapping training data, so a permutation test would be more appropriate than a t-test.

Thank you for the critical comment and the constructive suggestion.

We agree with the reviewer that the correlations resulting from the cross-validation were not strictly
independent because they used overlapping training data. This violated the assumption for applying t-
test to the (z-transformed) correlations. According to the reviewer’s suggestion, we used a permutation

test as a non-parametric alternative to the t-test. Despite the different methods for testing the statistical
significance, the results were very similar, highlighting the same set of areas that constituted the
semantic system. As below, we described our permutation test.

For this revision, we used 10-fold cross-validation by holding out 1/10 of the training data for validation
and repeating 10 times such that every sample in the training data had been used once and only once as
the validation data. The model was trained with the remaining 9/10 training data and validated against
the held-out data. The cross-validation accuracy was calculated separately for every voxel as the average
voxel-wise correlation between the measured and model-predicted responses in the validation data. We
evaluated the p value of the cross-validation accuracy by comparing it against a null distribution
obtained with block-wise permutation of the training data (Adolf et al., 2014). Specifically, we divided
the training stories into blocks (each including 20 seconds) and kept the feature time series intact within
each block but shuffled them randomly across blocks. Such block-wise shuffling disrupted the
relationship between the (shuffled) feature time series and the (unshuffled) fMRI time series. When the
encoding model was trained with such shuffled training data and then validated against the held-out
validation data, the cross-validation accuracy was a sample of permutation. By repeating the block-wise
shuffling for 100,000 times, we obtained 100,000 samples of permutation and formed a null
distribution.

**Figure 2** highlights the areas where voxels were statistically significant given the above permutation test
with FDR $q < 0.05$. We used these areas as a cortical mask and confined other analyses to this mask
(**Figure 4, 5, 6, 7, 8, Supplementary Figure 9**).

Accordingly, we have made the following revision to the main text.

From line 78 to 80 in Results:

*“By 10-fold cross-validation [25], the model-predicted response was significantly correlated with the*
*measured fMRI response (block-wise permutation test, false discovery rate or FDR $q < 0.05$) for voxels*
*broadly distributed on the cortex (Fig. 2).”*

From line 469 to 477 in Methods:

*“We further tested the statistical significance of 10-fold cross-validation for every voxel based on a block-*
*wise permutation test [57]. Specifically, we divided the training data into blocks; each block had a 20-sec*
*duration. We kept the HRF-convolved word features intact within each block but randomly shuffled the*
*block sequence for each of 100,000 trials of permutation. Before or after the block-wise shuffling, the*
*word feature time series had the nearly identical magnitude spectrum, whereas the shuffling disrupted*
*any word-response correspondence. For every trial of permutation, we ran the 10-fold cross-validation as*
*aforementioned, resulting in a null distribution that included 100,000 cross-validation accuracies with*
*permuted data. Against this null distribution, we compared the cross validation without permutation*
*and calculated the one-sided p-value while testing the significance with FDR $q < 0.05$.”*

L398. Unclear why a paired t-test was used here instead.

Thank you for the question.

In the revised manuscript, this question is no longer of concern. According to a related comment from
Reviewer #1, we have replaced the paired t-test with a paired permutation test, as explained below.

For each time of permutation, we randomly flipped the direction of every word pair. For example, the
relation vector of <book, pages> could be the vector difference between “book” and “pages” or
between “pages” and book” with equal probability. For a given semantic relation of interest, we mapped
the randomly flipped relation vector of every example word pair and averaged the resulting cortical
representation across word pairs. By repeating the permutation for N=100,000 times, we generated a
null distribution at every voxel. Against this null distribution, we compared the cortical representation of
interest obtained with intact word pairs (without any flipping). We calculated the probability that the
samples in the null distribution exceeded the cortical representation of interest and chose the significant
voxels with FDR $q < 0.05$. Based on this non-parametric statistic test, we have updated our figures (Figure
6, Figure 7, Supplementary Figure 9). For each semantic relation, the number of significant voxels is
listed in Supplementary Table 4.

Accordingly, we have made the following revision to the main text.

From line 152 to 154 in Results:

*“By using the encoding model, we mapped the pairwise word relationship onto voxels in the semantic
system (as shown in Fig. 2), averaged the results across pairs, and highlighted the significant voxels
(paired permutation test, FDR $q < 0.05$).”*

From line 565 to 571 in Methods:

*“For the averaged cortical projection, we tested the statistical significance for every voxel based on a
paired permutation test. In this test, we flipped every word pair at random for 100,000 trials. For every
trial, we calculated the model-projected cortical pattern averaged across the randomly flipped word
pairs, yielding a null distribution per voxel. Against this voxel-wised null distribution, we compared the
average voxel value projected from non-flipped word pairs and calculated the two-sided p-value with the
significance level at FDR $q < 0.05$. The resulting pattern of significant voxels was expected to report the
primary cortical representation of each semantic relation of interest.”*

L528: Data availability, the authors write “The imaging datasets and processing codes will be publicly
available later.” – This is a bit vague, and I believe a more concrete statement is warranted (e.g., specify
the number of months after publication). Also, at least the code should be made available for review.

Thanks for the comment. We support data sharing and commit to making the data and code available 3
456 months after the paper is accepted for publication, allowing anyone to review our methods and reuse
our data for further investigation. In addition, we are also making the major code available for review.

The statement about data availability has been revised accordingly.

From line 589 to 590 in Data availability:

*“The fMRI data, audio story stimuli, and word/word-pair samples that were used in this study will be
available within 3 months following the publication of this paper.”*

From line 592 to 593 in Code availability:

*“Codes for model training, testing, and validation will be released within 3 months following the*
*publication of this paper.”*

Reviewer #3 (Remarks to the Author):

Review of “Connecting concepts in the brain: Mapping cortical representations of semantic relations”
for Nature Communications (double-blind peer-review)

SUMMARY

This paper describes an fMRI study aimed at mapping conceptual relationships (e.g., “whole-part”) onto
cortical networks. Nineteen participants each listened to audio stories from The Moth Radio Hour,
including several stories that were unique to each participant (for training an encoding model) along
with one story that was presented to all participants (for testing the model). A semantic space was
based on a pre-trained word2vec model of word embeddings that represents words as 300-dimension
vectors of semantic features, and a voxel-wise encoding model related this semantic space to the fMRI
data by expressing each voxel’s response as a weighted sum of semantic features. This resulted in
whole-brain maps of regression coefficients corresponding to each concept. Within these voxelwise
maps, the left hemisphere appeared to be generally more selective for concrete concepts, while the
right hemisphere appeared to be more selective for abstract concepts. From there, the authors
operationalized conceptual relationships as the average of voxel-wise subtractions between whole-brain
maps corresponding to pairs of individual words. While a few intuitive conceptual relationships
surprisingly did not correspond to voxelwise differences anywhere in the brain, most contrasts revealed
significantly positive or negative to z-statistics widely distributed throughout the brain. The authors
interpret these differences as anti-correlated cortical networks and observe that several of the contrasts
appear to distinguish between areas generally associated with either the default mode network or the
frontoparietal attention network.

Thank you for the summary of our work. Given this summary, we believe we and the reviewer are on
the same page about the research strategy and findings in this study.

This is a very interesting and creative marriage of computational and linguistic models with fMRI data
that has the potential to provide new and important insights into neural representation. However, I
have several concerns regarding the analysis approach and the interpretation of findings that limit my
enthusiasm for the manuscript in its current form.

Thank you for the positive opinion about our research strategy.

According to the reviewer’s comments and suggestions, we have clarified, refined, or strengthened our
analysis approach as well as the resulting findings. The revised manuscript is much improved and
hopefully would lead to more enthusiastic evaluation.

Our reply to each comment is elaborated as below.

CONCERNS

1) Content and participant are confounded in training data.

We understand the reviewer's concern. In the original manuscript, we did not elaborate well how we
chose different stories for different subjects. In our study design, we counterbalanced the story stimuli
across subjects. By counterbalancing, the stories for different subjects similarly sampled the semantic
space (Supplementary Figure 1), the semantic categories of interest (Supplementary Figure 3), the
semantic relations of interest (Supplementary Figure 4), and a common set of frequently used words
(Supplementary Figure 2 and Supplementary Table 1).

Although we could not rule out the potentially confounding relationship between content and
participant, we expected this confound to be negligible and unlikely to invalidate our findings. To
support this argument, we have provided additional results and discussions as to how we chose and
counterbalanced the story stimuli for different participants.

See our following replies to the related comments.

To train the voxelwise encoding model, data was concatenated across participants who each listened to
different sets of stories about diverse topics. Specifically, each of the 19 participants contributed about
18 minutes of unique stories to the combined 5 hours 33 minutes of training data. For testing, every
subject listened to the same 7-minute story. I appreciate that by having participants listen to different
stories and then combining data across participants, the authors were able to more thoroughly sample
many more unique words than they would have been able to if each participant listened to the same set
of audio stories.

Thanks for your understanding and appreciation of our study design. Indeed, having different subjects
listen to different stories was central to this study because it allowed us to more thoroughly sample the
semantic space for proper training of the encoding model.

In addition, we would like to highlight two rationales for our design.

First, naturalistic stimuli, e.g. audio stories, are known to elicit cortical responses highly reproducible
across subjects (Hasson et al., 2010). When fMRI scans were acquired from different subjects listening to
different story stimuli, it was reasonable to expect that the measured activity from different brains
would be similar to the activity as if measured from the same brain. This assumption has been explicitly
used to map cortical activations with naturalistic audio or visual stimuli by evaluating the inter-subject
reproducibility of the voxel-wise signal during the same stimuli (Kauppi et al., 2010; Lerner et al., 2011).

Second, a large amount data is required to train the encoding model, which projects hundreds of
features to thousands of voxels. It is logistically challenging to acquire enough data from a single subject,
since a typical fMRI experiment lasts <2 hours per subject to avoid fatigue, which would compromise
data quality or confound interpretation. Balancing these considerations, having different subjects listen
to different stories is a good and realistic strategy. It allows us to combine data across subjects so as to
cumulatively gather enough data samples for training the encoding model without overfitting.

Accordingly, we have described our rationale for using different stories across different subjects.

From line 340 to 343 in Discussion:

*“In this study, we used different stories for different subjects to collect a large set (47,356 words) of*
*stimulus-response samples for training the encoding model. It is logistically difficult to acquire enough*
*data from a single subject. A typical fMRI experiment lasts <2 hours to avoid fatigue, while >11 hours of*
*fMRI scans as needed for the desired sample size would be too long to be realistic.”*

From line 392 to 394 in Methods

*“This design provided a large number of stimulus-response samples necessary to train the encoding*
*model for mapping hundreds of semantic features to thousands of cortical voxels.”*

Please also see our further replies to other related questions.

However, even though the model was always tested based on data from the same story, I am concerned
that individual differences (e.g., in laterality, which can be quite variable across participants) are
confounded with the words used to train the model.

We understand the concern that “individual differences are confounded with the words used to train
the model”. We admit that individual differences exist, e.g. laterality (Knecht et al., 2000; Gotts et al.,
2013). The participants were all right-handed, young, and native English speakers. So individual variation
is expected to be lesser compared to the general population. In addition, our design and results suggest
that individual variation is not a major confounding factor of concern.

In the revised manuscript, we have added discussion about individual variation.

From line 343 to 351:

*“This study design might be of potential concern that individual differences, e.g. laterality [26, 53], are*
*confounded with the words used for model training. If a subset of subjects is over-represented for one*
*semantic dimension and a different subset is over-represented for a different dimension, the trained*
*encoding model would reflect the idiosyncratic variation across individuals. To avoid this concern, we had*
*counterbalanced the stories across subjects. By counterbalancing, the stories for different subjects*
*similarly sampled the semantic space (Supplementary Figure 1), the semantic categories or relations of*
*interest (Supplementary Figure 3 & 4), and a common set of frequently used words (Supplementary*
*Figure 2).”*

See our following replies to other related comments.

For instance, Table S2 displays the number of paired samples for each semantic relationship (ranging
from 44 to 178). Ideally, each semantic relationship was evenly sampled across participants. Otherwise,
if a subset of participants is overrepresented for one semantic dimension, and a different subset of
participants is overrepresented for a different semantic dimension, then the trained encoding model will
reflect the individual differences of the participants.

We understand the reviewer’s concern. As the reviewer mentioned, in a hypothetical scenario, “if a
subset of participants is overrepresented for one semantic dimension, and a different subset of
participants is overrepresented for a different semantic dimension, then the trained encoding model will
reflect the individual differences of the participants.” If this scenario indeed happened, it could have
complicated the interpretation of our results concerning the model-predicted representations of
semantic categories or relations. However, this scenario did not happen in our study. In fact, we did not
find any evidence that the word samples were biased or overrepresented for one semantic dimension or
domain for any subset of participants.

We did counterbalance the different stories for different subjects. See Supplementary Method 1.

Although they were intended to include different sets of words, the stories used for different subjects
sampled the semantic space with similar distributions. To better illustrate this point, we visualized the
word samples for each subject in a 2-dimensional plane, which resulted from nonlinear dimension
reduction of the embedding space by using the t-Distributed Stochastic Neighbor Embedding (t-SNE)
algorithm (Maaten and Hinton, 2008) – a widely used method readily implemented as a built-in function
in MATLAB. The word samples for every subject were uniformly distributed and centered at the origin,
with similar distributions across subjects (Supplementary Figure 1). Therefore, the words used for
different subjects had similar distributions without any sampling bias noticeable for any subject.

Further, we examined how each semantic category was sampled across subjects, as suggested by the
reviewer. For example, the training data included 484 unique words in the “place” category and these
words occurred 5,975 times in total. Then we counted how many times those words occurred in the
stories used for each subject and plotted the resulting word counts by subject in a pie chart. As shown in
Supplementary Figure 3, the numbers of word samples in each category were comparable across
subjects. In other words, each category was close to being evenly sampled across subjects, while strictly
even sampling would be difficult to implement in realty with natural stories.

Similarly, we examined how each semantic relation was sampled across subjects. As shown in
Supplementary Figure 4, the sampling of semantic relation was roughly even across subjects, despite
slightly more variation than the sampling of semantic category. Although strictly even sampling across
subjects is ideal, it would be very difficult, if not impossible, to implement in realty with natural stories.

In addition, we also checked the frequently appeared words for each subject (Supplementary Figure 2
and Supplementary Table 1). Since each story we chose was a natural story told by a storyteller during a
live talk, the word appeared most was “I”. The top 50 words that were sampled in the whole training
dataset (Supplementary Figure 2) were consistent with the frequently used words in our daily life,
including “I”, “the”, “that”, “you”, “so”, “like” etc.

In short, we counterbalanced the stories for different subjects such that each semantic category or
relation of interest was close to being evenly sampled across subjects.

Notably, this is a different approach than in previous studies that the authors cite such as Huth et al.
(2016, Nature). In that study, each participant listened to the same 10 audio stories for model
estimation. Along with providing an unbiased training set, exposing participants to the same stimuli

allowed previous authors to separate variance that was consistent across participants from the
idiosyncratic variance of each participant (e.g., as displayed in Extended Data Figure 2).

Thank you for the comment. As the reviewer mentioned, our study design was different from previous
study from (Huth et al., 2016), we could not apply to our data the exactly same analysis as used in their
study. However, we could evaluate the individual variation in cortical activation by using the repeated
measures given the same testing story for every subject, as described in Supplementary Method 2.

*“Individual variation is unavoidable. For example, individuals vary in terms of laterality as demonstrated
before [2, 3]. In this study, we attempted to identify where individual variation was most pronounced in
the brain in terms of cortical activation with naturalistic audio stories. For the purpose, we used the fMRI
data when every subject was listening to the same testing story in two repeated sessions. For each
session, we split the data into 4 non-overlapping segments (or sub-stories). For each segment, we first
computed the voxel-wise correlation between the fMRI signals from the 2 repeated sessions in every
subject. As a result, the z-transformed correlation at every voxel depended on two factors: subject &
stimulus. We performed a two-way analysis of variance (ANOVA) and identified the voxels where the
factor of subject had a significant effect. In this way, we identified the cortical locations that have
significantly different involvement across subjects during natural speech comprehension.*

*As shown in Supplementary Fig. 11, the effect of individual variation (with uncorrected $p < 0.01$) was
found primarily in the auditory association cortex; however, the effect was largely diminished after
correction for multiple comparison (FDR $q < 0.05$). These results suggest that there was a minor to modest
level of individual variation in cortical activation with audio stories, consistent with the findings from
prior studies [4].”*

Accordingly, we have also discussed about individual variation in “Potential limitations and future
directions” in the revised manuscript.

From line 351 to 354 in Discussion:

*“In addition, the use of audio stories as naturalistic stimuli gave rise to highly reproducible cortical
responses across subjects, as shown in prior studies [20] and reinforced by our results (Supplementary
Fig. 11). See Supplementary Method 2 for more details on testing effects of individual variance.”*

With enough data, the appropriate counter balancing and the necessary measure checks, confounds
from training the model in this way could potentially wash out. However, there is currently no discussion
of how the authors controlled for this. The authors are not specific about how they selected the stimuli,
other than noting that the audio stories were intentionally of diverse topics.

Thank you for the comment. We apologize for having not elaborated how we controlled and
counterbalanced the story stimuli for different subjects. As we have mentioned in our replies to other
related comments, we agree with the reviewer that the content difference might be confounded by the
individual difference. However, we believe this potential confound is not of a major concern, given our
design and results.

We would like to refer the reviewer to the following revision in the main text.

From line 70 to 74 in Results:

*“By counterbalancing the stories across subjects, we sampled different words with different subjects,*
*such that the sampled words for every subject covered similar distributions in the semantic space*
*(Supplementary Fig. 1) and included a common set of frequent words (Supplementary Fig. 2 and Table 1),*
*while every semantic category or relation of interest was sampled roughly evenly across subjects*
*(Supplementary Fig. 3 & 4).”*

From line 396 to 405 in Methods:

*“In an attempt to sample a sufficiently large number of words in the semantic space, we intentionally*
*chose audio stories of diverse contents. Since different subjects listened to distinct (training) stories, we*
*further counterbalanced the stories across subjects. For every subject, the stories included different*
*words ($2,492 \pm 423$) but sampled similar distributions in the semantic space (Supplementary Fig. 1) [54].*
*For each semantic category or relation of interest, the associated words were roughly evenly sampled*
*across subjects (Supplementary Fig. 3 & 4). The stories presented to each subject also included a set of*
*common words used frequently in daily life (Supplementary Fig. 2). In total, the training stories include*
*5,228 unique words. By counterbalancing the stories across subjects, we attempted to avoid any notable*
*sampling bias that could significantly confound the idiosyncratic variation across subjects with the*
*variation of the sampled words across subjects. See Supplementary Method 1 for more details.”*

We have added Supplementary Method 1 to provide further details.

2) Intuitive semantic relations (e.g., “similar”) do not appear to map onto any informative voxels despite
adequate power.

The authors note that “not all human-defined semantic relations have an underlying vector field that
can be represented as a consistent cortical region or network” (p. 12). Indeed, in Table S2, the semantic
relations “similar” and “contrast” each map onto 0 voxels, and the relation “cause-effect” maps onto
just 11 voxels (compared to >12,000 voxels for several of the other semantic relations like “class-
inclusion”). This is puzzling to me, especially since the dataset includes more word pairs for “similar” and
“contrast” than for most other relations. It would be helpful for the authors to determine (or at least
more specifically speculate) why not all relations from the SemEval-2012 dataset map onto cortical
networks. I can imagine a post-hoc explanation for why there might be 0 reliable voxels for “similar”
(e.g., “house” vs. “home”). For instance, the cortical representations of “house” and “home” could be so
nearly identical that differences between these brain states are negligible.

Thank you for this interesting question. It is indeed counterintuitive that some intuitive relations (e.g.
“similar”) do not appear to map onto any informative voxels despite adequate power. We also agree
with the reviewer’s post hoc explanation that the cortical representations of two “similar” words may
have nearly identical cortical representations that their differences may be negligible.

In addition, we have another explanation. The “similar” relation is a type of symmetric relation. For
example, <house, home> holds a similar relation, while <home, house> also holds a contrast relation.
Flipping the two words in a “similar” relation does not change the relation. In our analysis, we represent
the relation between a pair of words as their differential vector and map this differential vector to the
cortex through linear projection. Flipping the order of two words in a pair reversed the cortical
projection but does not change the symmetric relation. In other words, two opposite cortical patterns

are equally valid as the representation of one relation. With the paired permutation test, it is
unsurprising why no voxel would be significant.

However, theoretically, I have trouble imagining why there would also be zero significant voxels for
“contrast” (e.g., “hot” vs. “cold”). If the authors could convincingly explain why some intuitive
conceptual relationships are distinctly represented whereas other intuitive relationships are not
represented at all, this would be, in my opinion, the most interesting and important finding of the paper.
I am concerned, however, the dissociations reflect deficiencies and confounds in the model (e.g., as
described in my first concern) as opposed to new discoveries about the brain.

Like “similar”, “contrast” is also a symmetric relation. For example, <hot, cold> holds a contrast relation,
while <cold, hot> also holds a contrast relation. As explained above, this type of symmetric relation is
not expected to map onto any informative cortical representation, at least with our method.

In contrast to symmetric relations, such relations as “whole-part”, “object-attribute”, “class-inclusion”
etc. are asymmetric relations. The relation is directed such that flipping two words in a pair changes the
relation. Such directed and asymmetric relations are likely able to map onto a cortical pattern consistent
across word-pair samples. But not all asymmetric relations necessarily have cortical representations. For
example, “cause-effect” is quite intuitive but does not appear to have a clear cortical representation. It
is still puzzling and awaits future investigations. It is also reasonable that a nonsensical relation, e.g.
“object-nonattribute”, does not have any cortical representation (Supplementary Fig. 9).

Our method is more suitable for addressing asymmetric relations, instead of symmetric relation. We
agree with the reviewer that the lack of cortical representation for “similar” or “contrast” is very
interesting. However, we would like to be cautious and refrain from drawing a firm conclusion in this
paper but would take this as a hypothesis to be tested with future experiments. Within the scope of this
paper, the focus should remain as the cortical representations of “asymmetric” relations and semantic
categories.

In the revised manuscript, we have added related discussions about this as a potential limitation.

From line 355 to line 362 in Discussion:

*“Some intuitive semantic relations, e.g. “similar” and “contrast”, did not map onto any informative*
*voxels despite an adequate sample size (Supplementary Fig. 9). Although this might appear*
*counterintuitive, it is not surprising at all because such relations are both symmetric. For example, (“hot”,*
*“cold”) holds a “contrast” relation, while (“cold”, “hot”) also holds the same relation. Likewise, the*
*“similar” relation is also symmetric. In contrast, other relations, e.g. “whole-part”, and “case-relations”,*
*are asymmetric. The relation is directed such that flipping two words in a pair changes the relation. Since*
*we use differential vectors or representational contrast to evaluate word relationships, our method is*
*more suited for addressing asymmetric relations, instead of symmetric relations.”*

3) Exclusively voxelwise approach to examine distributed representations.

Ideas throughout the paper are focused on distributed neural representations. However, the encoding

model used to examine distributed neural representations is univariate (i.e. each voxel is separately
modeled), with hypotheses tested by counting or visualizing the number of individual voxels labeled by
each category or relation. Although terms such as “anticorrelated networks” suggest analysis of how
different parts of the brain interact in different contexts, analyses are limited to the selectivity of
individual voxels visualized as schematic maps of the brain and interpreted in terms hemispheric
asymmetries and the relative loading of feature dimensions.

Thank you for the critical comment. As the reviewer suggested, the results in the original analysis were
based on voxel-wise univariate analysis. Even though the results reveal a distributed cortical pattern
associated with a category or relation, it is more suitable to refer to those patterns as co-activation and
deactivation as opposed to “networks”.

In the revised manuscript, we describe the results by using the term of “co-occurring patterns of cortical
activation and deactivation”, and then discuss our interpretation of such patterns as anti-correlated
networks. In addition, we have extended our analysis by including multi-variate pattern analysis, as the
reviewer suggested. See our reply to the next comment.

Accordingly, we have made the following revision to the main text.

From line 160 to 164 in Results:

*“The co-activation and deactivation pattern indicated that conceptual progression from “part” to*
*“whole” manifested itself as increasing deactivation of FPN alongside increasing activation of DMN,*
*whereas progression from “whole” to “part” was shown as the reverse cortical pattern varying in the*
*opposite direction,...”*

From line 284 to 293 in Discussion:

*“Such cortical patterns often manifest themselves as co-occurring activation and deactivation of regions*
*(Fig. 6, 7 & 8). We interpret this co-activation and co-deactivation as an emerging pattern when the*
*brain relates two concepts that hold a meaningful relation, reflecting the progression from one concept*
*to the other. This pattern encodes generalizable differential relations between concepts, as opposed to*
*concepts themselves, serving differential coding that transcends any conceptual domain or category (Fig.*
*6). Speculatively, this network-based coding of semantic relation is an important mechanism that*
*supports analogical reasoning [45], e.g. matching similar relations with different word-pair samples [31].*
*This brain mechanism might further facilitate humans learning new concepts by connecting them to*
*existing concepts through established semantic relations.”*

From line 307 to 316 in Discussion:

*“We interpret the co-activation/deactivation patterns as “anti-correlated networks”. This interpretation*
*is reasonable given the notion of “activate together wire together”. Task-related patterns of cortical*
*activation resemble those emerging from spontaneous activity or resting state networks [46]. In the*
*context of semantics, the anti-correlated networks reported herein encode a semantic relation, or the*
*direction in which one concept relates to another. For example, conceptual progression from “part” to*
*“whole” has a cortical signature as co-occurring activation in DMN and deactivation in FPN (Fig. 6b). The*
*opposite direction from “whole” to “part” involves the same regions or networks but reverses their*
*polarity in terms of activation or deactivation. In this example, the cortical co-activation/deactivation*

*pattern is nearly identical to the anti-correlated networks observed with resting state fMRI [47], and*
*therefore it is likely to be intrinsic and supported by underlying structural connections.”*

Critically, using this univariate approach, the authors were not able to identify voxelwise differences for
seemingly basic and intuitive relationships like similarity and contrast. If the model is correct (but see my
first and second concerns above), it could be that these relationships are represented by patterns and
interactions across voxels and regions as opposed to the individual tuning properties of any one voxel or
region. In this case, widely used multivariate approaches to fMRI analysis like multivariate pattern
analysis and functional connectivity may be important to consider as well (and may also be more
consistent with ideas of representation). For instance, representational similarity analysis (e.g.,
Kriegeskorte et al., 2008 Frontiers) could also be applied to further compare between word2vec
representations and brain representations for intuitive and yet seemingly intractable semantic
relationships like similarity and contrast.

We thank the reviewer for this constructive suggestion. Univariate voxel-wise analysis is a valid method
to map distributed cortical representations of semantic categories or relations. However, we agree that
the voxel-wise analysis is not the only way or arguably not the most intuitive way to address cortical
networks, which are increasingly evaluated with bi-variate (e.g. correlation) or multivariate (e.g. PCA or
ICA) analyses.

The encoding model maps word relations to cortical patterns. In our original analysis, we focused on
separate analysis of individual voxels as opposed to the patterns. In line with what the reviewer
suggested, we added a multivariate analysis, as described in the revised Methods in the main text as
well as Supplementary Method 7.

From line 572 to 587 in Methods:

*“Complementary to the voxel-wise univariate analysis, we also applied a multivariate pattern analysis*
*(MVPA) to the cortical projection of word relations [59]. Unlike the univariate analysis, MVPA was able to*
*account for interactions between voxels and uncover likely multiple cortical patterns associated with*
*each semantic relation of interest. Specifically, given a class of semantic relation, we concatenated the*
*cortical pattern projected from every word-pair samples in that class and calculated a covariance matrix*
*describing the similarity of representations between samples [60, 61]. By using principal component*
*analysis (PCA), we obtained a set of orthogonal components (i.e. eigenvectors), each representing a*
*cortical pattern that accounted for the covariance to a decreasing extent. We chose the top-10 principal*
*components and calculated the pattern-wise cosine similarity between every component and the cortical*
*projection of every word-pair sample. For each component, we averaged the cosine similarity across all*
*samples of the given semantic relation and tested the statistical significance based on one-sample t-test*
*($p < 0.01$). Specifically, for any relation with multiple significant components, we grouped and sorted the*
*word-pairs based on their corresponding cosine similarities with each component. For each component,*
*we listed the top-4 word-pairs with the highest cosine similarity for intuitive understanding as to whether*
*the component was selective to a sub-class of that relation. See more details in Supplementary Method*
*7.”*

While being generally consistent with the results obtained with univariate analysis, the results obtained
with multivariate analysis revealed two significant cortical patterns associated with some semantic
relations, such as “object-attribute”, “case relations”, and “space-associated”.

To report this intriguing result, we have added Fig. 8 to the main text, as well as Supplementary Fig. 9.

When there were two patterns representing one relation, it appeared that each pattern was selectively
associated with a sub-class of the relation. This might be because a semantic relation defined by human
intuition does not exactly match the relation as represented by the brain. We added this result and our
interpretation to the revised manuscript.

From line 181 to 195 in Results:

*“The voxel-wise univariate analysis restricted the representation of each semantic relation to one cortical*
*pattern while ignoring the interactions across voxels and regions. This limitation led us to use a principal*
*component analysis to decompose the cortical projection of the difference between every pair of words*
*in each semantic relation. This multivariate analysis revealed two cortical patterns that were statistically*
*significant (one-sample t-test, $p < 0.01$) in representing the semantic relation of “object-attribute”, “case*
*relations”, or “space-associated”, but only revealed one pattern for the relation of “whole-part”, “class-*
*inclusion”, “time-associate”, or “cause-effects” (Supplementary Fig. 9). Interestingly, when two cortical*
*patterns represented the same semantic relation, they seemed to correspond to different sub-classes of*
*that relation (Fig. 8). For the relation of “object-attribute”, one cortical pattern corresponded to*
*“inanimate object-attribute”, e.g. (“candy”, “sweet”), and the other corresponded to “human-attribute”,*
*e.g. (“coward”, “fear”) (Fig. 8a). Similarly, the two cortical patterns for “case relations” corresponded to*
*“agent-instrument” and “action-recipient”, respectively (Fig. 8b). The “space-associated” relation was*
*distinctively represented for its two sub-classes: “space-associated item” and “space-associated activity”*
*(Fig. 8c). The cortical patterns that represented a relation, as obtained with either the multivariate or*
*univariate analysis, highlight generally similar regions (Supplementary Fig. 9).”*

From line 294 to 298 in Discussion:

*“A semantic relation as defined by human intuition may not exactly match the relation as represented by*
*the brain. A nominal relation may be heterogeneous and contain multiple sub-classes each being*
*represented by a distinct cortical pattern. Results obtained with multivariate pattern analysis support*
*this notion (Fig. 8). It is also reasonable that a nonsensical relation, e.g. “object-nonattribute”, does not*
*have any cortical representation (Supplementary Fig. 9).”*

4) ROI model fits displayed in Figure 1 appear to be inconsistent or circular with whole-brain statistics.

To show that the word2vec-based encoding model was able to capture the quantitative and
generalizable relationships between cortical responses and word attributes the stimulus set, the authors
display overlays of the BOLD timeseries and model predictions for various regions (Fig 1b). However,
these overlays appear to depict much higher correlations than suggested by the whole-brain statistics
displayed in Fig 1a. For instance, the first pair of lines representing IFG suggests to me an exceptionally
strong correlation between the signal and model prediction. Yet, the color scale for Fig 1a ranges from
0.3 to 0.6, with most of both left and right IFG completely masked out (i.e., a correlation < 0.3). That is,

right IFG is almost entirely empty and left IFG also appears to be mostly empty with the exception of a
very anterior cluster bordering IFG. This suggests that the signal timeseries (and its relationship to the
model prediction) could not possibly reflect IFG as it would be anatomically defined. Possibly, the signal
timeseries specifically reflects the significant cluster of voxels that are within proximity to left IFG. If so,
“double dipping” gives the impression that the model is much more accurate than it actually is. While
I’m sensitive to circular inference in neuroimaging, this might otherwise be a minor concern because it
can be easily addressed and the authors do not actually report the corresponding statistics. However,
because the model fits are so prominently displayed in the first figure and provided as the primary
evidence that the model is highly accurate, they strongly influence the reader’s interpretation of
subsequent results.

Thank you for the critical comment. We apologize for not describing the results with sufficient details;
As a result, our intention of time series comparison could be mistaken.

In the original manuscript, the time series shown in the original Fig. 1b were the measured and
predicted signals at a single “best-example” voxel within a ROI. Our intention is to demonstrate that
different regions (or locations) have highly distinct response time series given the same stimuli, and that
the different response time series are predictable by the model. However, we agree with the reviewer
that “cherry-picking” an example is not an optimal way for result reporting. To avoid “double dipping”,
we have now used predefined ROIs (Fan et al., 2016) and reported the ROI-averaged response time
series. See Fig. 3c in the revised manuscript. Specifically, we have shown the time series from 16 ROIs,
including the left and right IFS, SMG, AG, STG, MT, FuG, PhG, and PCC, as shown in Fig. 3b. For each ROI,
the corresponding statistics regarding the location of the ROI, size of the ROI, and correlation values
within the ROI are listed in Supplementary Table 2.

Accordingly, we have made the following revision to the main text.

From line 89 to 98 in Results:

*“As shown in Fig. 3a, we found that the encoding model was able to reliably predict the evoked*
*responses in the inferior frontal sulcus (IFS), supramarginal gyrus (SMG), angular gyrus (AG), superior*
*temporal gyrus (STG), middle temporal visual area (MT), left fusiform gyrus (FuG), left parahippocampal*
*gyrus (PhG), and posterior cingulate cortex (PCC) (block-wise permutation test, FDR $q < 0.05$). These*
*regions of interest (ROIs), as predefined in the human brainnetome atlas [27] (Fig. 3b, see details in*
*Supplementary Table 2), showed different response dynamics given the same story, suggesting their*
*highly distinctive roles in semantic processing (Fig. 3c). Despite such differences across regions, the*
*encoding model was found to successfully predict the response time series averaged within every ROI*
*except the right FuG (Fig. 3c), suggesting its ability to account for the differential semantic coding (i.e.*
*stimulus-response relationship) at different regions.”*

From line 500 to 504 in Methods:

*“In addition, we extracted the fMRI responses at ROIs predefined in the Human Brainnetome Atlas, which*
*is a connectivity-based parcellation reported in an independent study [27]. We averaged the measured*
*and model-predicted fMRI responses within each given ROI, and compared them as time series (see Fig.*
*3c). The corresponding statistics regarding the location, size, and prediction performance of each ROI are*
*listed in Supplementary Table 2.”*

Reference

- Adolf D, Weston S, Baecke S, Luchtman M, Bernarding J, Kropf S (2014) Increasing the reliability of data
analysis of functional magnetic resonance imaging by applying a new blockwise permutation
method. *Frontiers in neuroinformatics* 8:72.
- Brysbaert M, Warriner AB, Kuperman V (2014) Concreteness ratings for 40 thousand generally known
English word lemmas. *Behavior research methods* 46:904-911.
- Fan L, Li H, Zhuo J, Zhang Y, Wang J, Chen L, Yang Z, Chu C, Xie S, Laird AR (2016) The human
brainnetome atlas: a new brain atlas based on connectonal architecture. *Cerebral cortex*
26:3508-3526.
- Gotts SJ, Jo HJ, Wallace GL, Saad ZS, Cox RW, Martin A (2013) Two distinct forms of functional
lateralization in the human brain. *Proceedings of the National Academy of Sciences* 110:E3435-
E3444.
- Hasson U, Malach R, Heeger DJ (2010) Reliability of cortical activity during natural stimulation. *Trends in*
*cognitive sciences* 14:40-48.
- Huth AG, de Heer WA, Griffiths TL, Theunissen FE, Gallant JL (2016) Natural speech reveals the semantic
maps that tile human cerebral cortex. *Nature* 532:453.
- Kauppi J-P, Jääskeläinen IP, Sams M, Tohka J (2010) Inter-subject correlation of brain hemodynamic
responses during watching a movie: localization in space and frequency. *Frontiers in*
*neuroinformatics* 4:5.
- Knecht S, Deppe M, Dräger B, Bobe L, Lohmann H, Ringelstein E-B, Henningsen H (2000) Language
lateralization in healthy right-handers. *Brain* 123:74-81.
- Lerner Y, Honey CJ, Silbert LJ, Hasson U (2011) Topographic mapping of a hierarchy of temporal
receptive windows using a narrated story. *Journal of Neuroscience* 31:2906-2915.
- Maaten Lvd, Hinton G (2008) Visualizing data using t-SNE. *Journal of machine learning research* 9:2579-
2605.
- Miller G (1998) *WordNet: An electronic lexical database*: MIT press.

Reviewers' Comments:

Reviewer #2:

Remarks to the Author:

The authors have addressed all my concerns

Reviewer #3:

Remarks to the Author:

The authors have sufficiently addressed each of my concerns.